🔓 | **Open Peer Review** | Host-Microbial Interactions | Research Article

# Interaction of host gene-gut microbiota in male grading of *Macrobrachium rosenbergii*

Xiuxin Zhao,[1] Miuying Cai,[2] Shunkai Yin,[1] Ziqi Zhou,[1] Jie Yang,[1] Yuqing Shen,[1] Zhenglong Xia,[2] Qiongying Tang,[1] Guoliang Yang,[1,2] Shaokui Yi,[1] Quanxin Gao[1]

**ABSTRACT** The giant freshwater prawn (GFP; *Macrobrachium rosenbergii*), a crustacean of high nutritional and economic value, is crucial for aquaculture. During the same growth cycle, male GFPs develop into three distinct forms: small males, orange claw males, and blue claw males. These morphotypes display varying social behaviors, which severely constrain their industrial development. To address this, this study collected male GFP samples at critical developmental time points (100, 110, and 120 days post-hatching) for phenotypic trait measurement and analysis to obtain external morphological data. Through gut microbiota diversity analysis, we identified key gut bacteria (*Lactococcus garvieae* and *Lactobacillus taiwanensis*) influencing male morphotype differentiation. Transcriptomic analysis revealed host Kyoto Encyclopedia of Gene and Genome pathways and key genes (*Wnt-6*, *CTSB*, *CTSL*, *PPAE*, and *TP53*) associated with morphotype differentiation. The interactions among phenotypic traits, gut microbiota, and key genes were systematically studied through association analysis. Weighted gene co-expression network analysis was employed to construct co-expression modules, from which critical gene modules influencing phenotypic variation were identified. Through association network analysis, we established an "*Achromobacter*-CD-TRINITY_DN93139_c0_g2 (calpain clp-1)" interaction model. Our findings provide novel insights into the genetic enhancement of GFPs and offer guidelines for future research regarding gut symbiotic bacteria and breeding initiatives.

**IMPORTANCE** Male *Macrobrachium rosenbergii* (giant freshwater prawn [GFP]) in the same growth cycle will develop into small males, orange claw males, and blue claw males. This individual heterogeneity in growth significantly impacts the benefits of aquaculture. However, the factors influencing the differentiation of male GFP morphotype remain unclear. This study analyzed the phenotypic data of various GFP levels, the structure of the intestinal microbiota, and the differential genes within the gonadal transcriptome at critical time points of male GFP-level type differentiation. The aim was to explore the potential role of intestinal microbiota and differential genes in this phenomenon. This study offers new insights into the research on the phenomenon of male GFP-level type differentiation.

**KEYWORDS** *Macrobrachium rosenbergii*, morphotype differentiation, phenotypic traits, gut microbiota, transcriptome

T he giant freshwater prawn (GFP; *Macrobrachium rosenbergii*), a member of the family Palaemonidae within the order Decapoda (1), is often referred to as the "king of freshwater prawns." This species is known for its rapid growth and short cultivation cycle (2–5). It is predominantly distributed in tropical and subtropical regions of the Pacific. The GFP is a crucial aquaculture species in many Asia-Pacific countries and holds significant economic value (6). Since its introduction to China in the 1970s, GFP

**Peer Reviewers** Shimming Peng, East China Sea Fisheries Research Institute, Shanghai, China; Chenghui Wang, Shanghai Ocean University, Shanghai, China

Address correspondence to Quanxin Gao, gaoqx2008@163.com, or Shaokui Yi, yishaokui@foxmail.com.

Xiuxin Zhao and Miuying Cai contributed equally to this article. The author order was determined based on the contribution to the article.

The authors declare no conflict of interest.

See the funding table on p. 19.

aquaculture has expanded significantly, making China the largest global producer (7). In 2024, the global production of *Macrobrachium rosenbergii* is approximately 356,000 tons, with China's total output exceeding 196,000 tons (8). The GFP exhibits pronounced sexual dimorphism, with significant differences between its male and female individuals (9). After approximately 21 days of metamorphic development, the GFP larvae enter the post-larval stage, during which females exhibit relatively uniform growth, while males display marked developmental variability (10). This phenomenon, termed heterogeneous individual growth, is characterized by the differentiation of male individuals into three distinct morphotypes within the same growth cycle: small males (SMs), orange claw males (OCs), and blue claw males (BCs) (Fig. 1A) (11). These morphotypes differ significantly in morphological traits, reproductive behavior, social hierarchy, and growth rates (12, 13). The social hierarchy of these three male morphotypes is as follows: BC > OC > SM (14, 15). SMs are characterized by transparent or pink claws and exhibit slow growth; OCs have medium-sized orange claws and grow rapidly, while BCs are larger, with elongated carapaces and robust claws (14, 16), although their growth rate is slower than that of OC (12, 13).

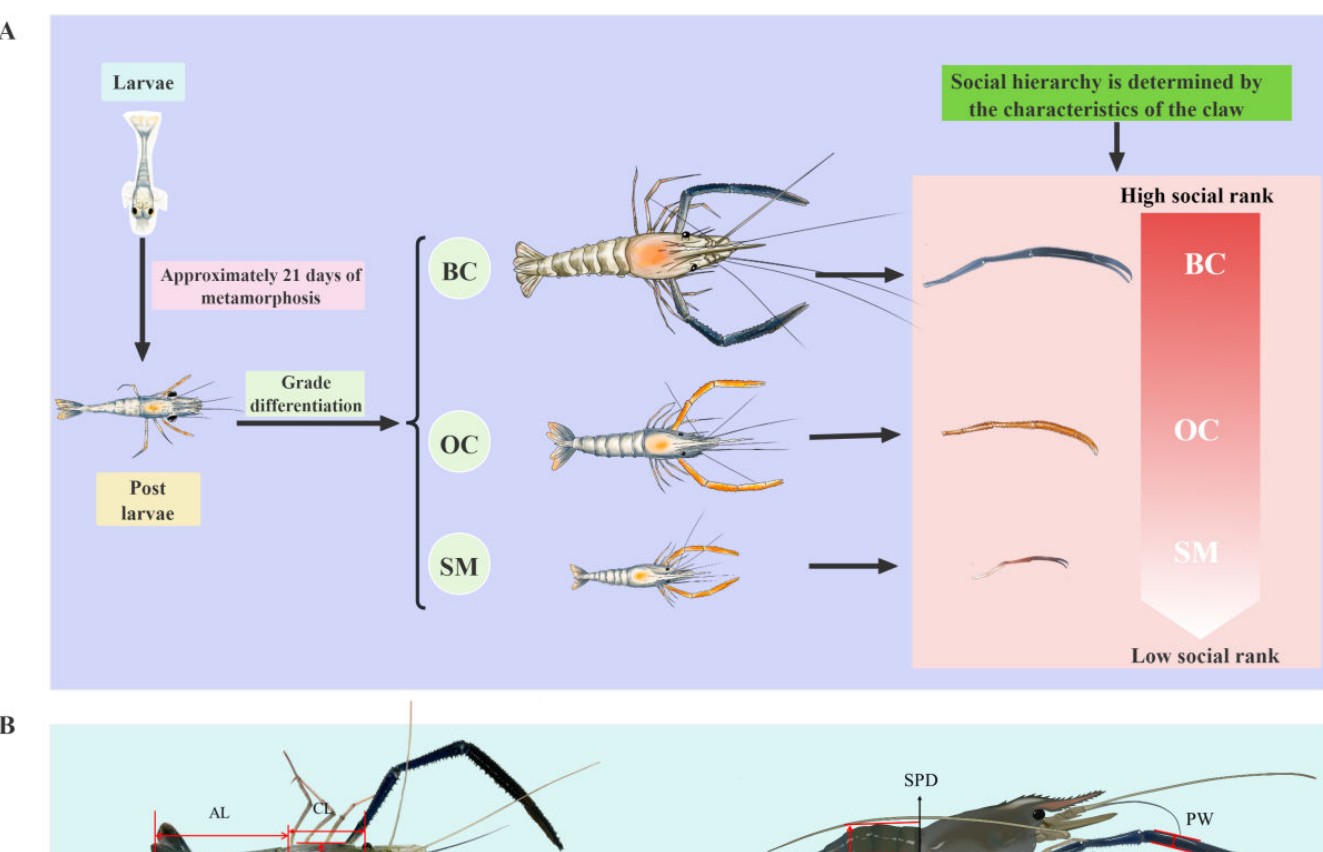

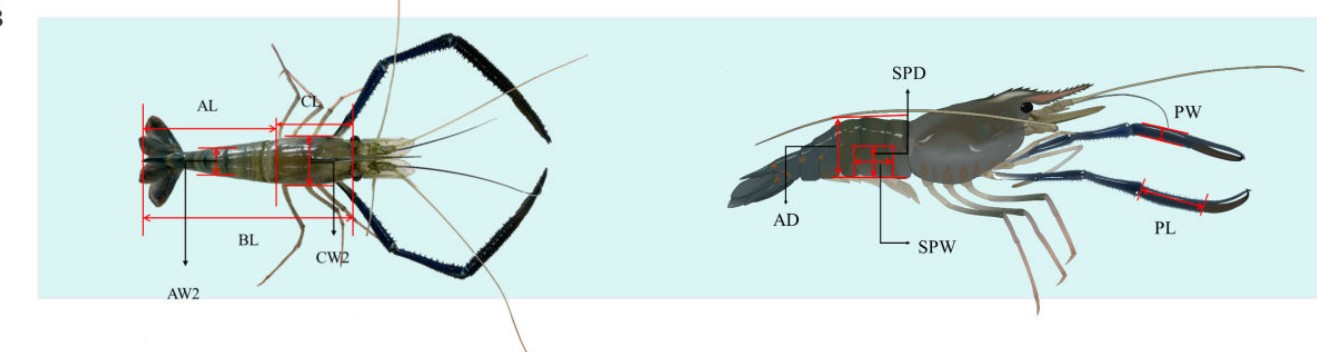

FIG 1 Developmental process of male giant freshwater prawn. (A) Development and morphotype differentiation of male GFP. Three different morphological types of male GFP: blue claw males, orange claw males, and small males. (B) Measurement of phenotypic traits of male GFP. BL, body length; CL, carapace length; AL, abdominal length; AW-2, abdominal width; CW-2, carapace width; AD, abdominal depth; CD, carapace depth; SPW, side panel width; SPD, side panel depth; PL, palm length; PW, palm width.

Despite these advances in understanding the basic biology and aquaculture potential of GFP, the factors influencing the differentiation of male GFP morphotype remain unclear. The concept of "hologenome" is gaining increasing attention for the investigation of microbiome-host relationships in aquatic organisms. Notably, the phenotypic traits of animals are not determined solely by the host genotype but are also influenced by the gut microbiota of the host (17). Gut microbiota, often referred to as "phenotype sculptors," play a pivotal role in the physiological development of the host (18). The gut microbiota even aids the host in adapting to changes in diet, environment, and developmental stages and, hence, significantly contributes to the formation of host phenotypes (19). All organisms are "holobionts," that is, a complex comprising the host and associated microorganisms (20, 21). The co-development, co-metabolism, and co-evolutionary relationships between the host and microorganisms jointly influence the traits of the host. The composition and abundance of gut microbial communities are regulated by the genetic factors of the host and, in conjunction with host genes, jointly influence the phenotypic traits of the host (22).

Furthermore, hosts selectively harbor specific microbial communities in their intestines, providing conditions for the proliferation and functionality of symbiotic bacteria, while gut microbiota, in turn, confer additional phenotypic traits to the host (23). It is well known that the anglerfish uses a fluorescent esca on its head to attract prey in dark ocean depths; its light organs contain dense, extracellular populations of bacteria that emit light through transparent tissue (Fig. S1) (24). It has recently reported that gut microbiota can regulate the social behaviors of aquatic animals, including common carps, zebrafish, and African cichlid fish (Fig. S1) (25–27). Studies have shown that the gut microbiota structure of GFPs undergoes dynamic changes during the growth and development of the host (28) and that a close relationship exists between gut microbiota and host morphological differentiation (29). In addition, a significant association between gut microbial communities and phenotypic traits of male GFP has been identified (30). This "host-gut microbiota" interaction affects food intake, metabolism, and body weight of the host, thereby giving rise to different social ranks among hosts (31). Since the different GFP males exhibit distinct social hierarchies and behaviors, we postulate that the gut microbiota can regulate the male grading and social behaviors of GFPs.

In male GFPs with identical genetic backgrounds, the variation in complex traits is not due to the presence or absence of specific genes but rather is a result of changes in gene regulatory mechanisms that determine when and where a gene or an entire regulatory module is expressed (32). The gonads play a crucial role in the development of crustaceans, participating in several key physiological processes, including hormone secretion, fertilization, and gametogenesis (33). As a representative aquatic species, GFP exhibits reversible gonadal development regulated by gene expression (34). Gonadal development is a critical phase in the GFP life cycle and is a significant reflection of its overall developmental status. Transcriptome sequencing can effectively reveal gene expression patterns in crustaceans (35). Therefore, the underlying mechanisms of male GFP morphotype differentiation can be elucidated by analyzing transcriptome sequencing data to identify potential functional genes influencing male GFP morphotype differentiation and correlating these findings to gut microbiota data.

Collectively, these considerations highlight the need for a comprehensive study to explore the complex interplay between phenotypic traits, gut microbiota, and genetic factors in male GFP morphotype differentiation. The morphotype differentiation of male GFP poses a major challenge to aquaculture productivity. Enhancing the proportion of rapidly growing morphotypes (OCs; BCs) while reducing the slow-growing SMs has become a central focus in global GFP breeding research. Previous studies have demonstrated significant correlations between phenotypic traits, gut microbiota, and genetic factors across male morphotypes (29). It is hypothesized that during male GFP development, host-gut microbiota interactions play critical roles in phenotypic divergence (30); however, the developmental dynamics of male morphotype

differentiation and associated shifts in gut microbiota remain poorly understood. To address this, this study aims to measure phenotypic traits in male GFPs (SMs, OCs, and BCs) from a single family (shared genetic background) at their key developmental stages. The study also analyzes the composition and diversity of the gut microbiota using 16S rRNA high-throughput sequencing coupled with bioinformatics approaches. Differential genes influencing morphotype differentiation are identified by integrating gut microbiota profiles, phenotypic data, and gonadal transcriptomics. The study also performs association analyses that further elucidate the tripartite relationships among gut microbiota, phenotypic traits, and gene expression, revealing molecular mechanisms underlying morphotype-specific phenotypic variation. The study findings provide critical insights for developmental studies and genetic improvement strategies aimed at optimizing GFP aquaculture sustainability.

## MATERIALS AND METHODS

### Sample collection

Giant freshwater prawns (*Macrobrachium rosenbergii*) of the "Shufeng No. 1" variety were obtained from Jiangsu Shufeng Prawn Breeding Co. Ltd (Gaoyou, Jiangsu, China). The selected male GFPs belonged to a single-family population, ensuring a uniform genetic background. All male individuals were of the same age and reared in the same pond under standardized conditions. The pond had an area of approximately 667 m² with a depth of 2–2.2 m. The water temperature of the pond was maintained at 28°C–32°C. Water quality parameters encompassed a pH range of 7.5–8. The GFPs were fed a specialized GFP diet provided by Jiangsu Fuyuda Grain Products Co., Ltd. Special feed contained no probiotics or antibiotics. The main components of the feed are crude protein ≥38%, crude fat ≥5%, and crude fiber ≤7%. The pond initially contained 7,000 post-larvae with a female-to-male ratio of 2:5. At 100, 110, and 120 days post-hatching, healthy male GFPs were classified based on their claw color, spine morphology, behavior, and growth characteristics (36). At each time point, we selected 66 individuals for sample collection (18 BCs, 18 OCs, and 30 SMs), and a total of 198 individuals were collected at the 3 time points. Due to the relatively small sample size of the SM-grade tissue, in order to ensure the required sample size for sampling, the number of SM samples was greater than that of the other phenotypes.

### Phenotypic trait measurement

After blotting the surface moisture on the GFPs with gauze, phenotypic traits of the males were measured following established protocols (30, 37, 38). An electronic balance (Mettler Toledo ME203E, precision: 0.01 g) and a digital vernier caliper (Mitutoyo CD-15CX, precision: 0.01 mm) were used to measure 15 phenotypic traits: total weight (TW), carapace weight (CW1), abdominal weight (AW1), weight of the second leg (WSL), body length (BL), carapace length (CL), carapace width (CW2), carapace depth (CD), palm length (PL), palm width (PW), side panel depth (SPD), side panel width (SPW), abdominal length (AL), abdominal width (AW2), and abdominal depth (AD). The measurement procedures are illustrated in Fig. 1B.

### Tissue sample collection

To minimize exogenous bacterial contamination, each individual was immersed in 75% ethanol for 30 seconds and then rinsed three times with sterile PBS. Whole intestine and gonadal samples were aseptically collected from SM, OC, and BC individuals using sterile scissors and forceps. Samples from three individuals were pooled to form one biological replicate, resulting in six replicates per group. After flash freezing in liquid nitrogen, all samples were stored at −80°C until further analysis. Six biological replicates were analyzed for 16S rRNA sequencing and gonadal transcriptomics.

## 16S rRNA sequencing

To investigate the structure and diversity of gut microbial communities in male GFPs at 100, 110, and 120 days post-hatching, high-throughput sequencing was performed on intestinal samples from different morphotypes. The V3-V4 hypervariable region (468 bp) of the 16S rRNA gene was amplified using specific primers: 338F (5′-ACTCCTACGGGAGGC AGCAG-3′) and 806R (5′-GGACTACHVGGGTWTCTAAT-3′). PCR amplification was followed by paired-end sequencing on an Illumina NovaSeq X Plus platform (Illumina, San Diego, CA, USA).

## Transcriptome analysis

Total RNA was extracted from gonadal tissue samples stored at −80°C using TRIzol reagent (TaKaRa Bio Inc., Dalian, China), following established protocols (37, 39). RNA purity and concentration were assessed using a Nanodrop 2000 spectrophotometer (Nanodrop Technologies, Wilmington, DE, USA) and a Qubit 2.0 fluorometer (Thermo Fisher Scientific, Waltham, MA, USA). RNA integrity was evaluated using an Agilent 5300 Fragment Analyzer (Agilent Technologies, Santa Clara, CA, USA) to determine the RNA quality number (RQN). Only high-quality RNA samples meeting the following criteria were used for cDNA library construction and transcriptome sequencing: total RNA ≥1 μg, concentration ≥30 ng/μL, RQN >6.5, and OD260/280 ratio between 1.8 and 2.2. The cDNA libraries were prepared using the TruSeq RNA Sample Preparation Kit (Illumina, San Diego, CA, USA), according to the manufacturer's instructions. Paired-end sequencing of the amplified products and transcriptome sequencing of the cDNA libraries were performed on the Illumina NovaSeq 6000 platform (Illumina, San Diego, CA, USA). Sequencing services were provided by Shanghai Majorbio Bio-Pharm Technology Co., Ltd. (Shanghai, China).

## Sequencing data quality control, transcriptome assembly, and gene functional annotation

After base calling using CASAVA, raw data were stored in the FASTQ format. To ensure data quality and reliability, low-quality reads containing poly-N sequences, reads shorter than 50 bp, and reads with an average sequence quality of below Q20 were removed. Clean data were *de novo* assembled using Trinity (https://github.com/trinityrnaseq/trinityrnaseq/wiki) (40). The initial assembled sequences were optimized and filtered using TransRate (http://hibberdlab.com/transrate/) (41) and CD-HIT (http://weizhongli-lab.org/cd-hit/) (42). The optimized sequences were evaluated for assembly quality using BUSCO (43). The refined transcripts were aligned against six major public databases: NCBI NR (non-redundant protein sequences), Swiss-Prot, Pfam (protein family), EggNOG (evolutionary genealogy of genes: non-supervised orthologous groups), GO (Gene Ontology), and KEGG (Kyoto Encyclopedia of Genes and Genomes). Functional annotations were statistically analyzed for each database. Gene and gene product functional classifications were determined based on GO annotations, while biological pathways of differentially expressed genes (DEGs) were elucidated using KEGG pathway annotations.

## Identification of DEGs and GO/KEGG enrichment analysis

DEGs were identified using the DESeq2 R package (v1.20.0) with the thresholds of $|log2(fold\ change)| > 1$ and $q$-value $< 0.05$ (44). GO and KEGG enrichment analyses of DEGs were performed using the GOSeq (v1.10.0) (GAP TECH, Beijing, China) and KOBAS (v2.0.12) (Center for Bioinformatics, Peking University, China) software packages, respectively. A false discovery rate <0.05 was considered statistically significant (45).

## Correlation analysis

Pearson correlation analysis was employed to evaluate the relationships among phenotypic traits, gut microbiota biomarkers, and DEGs. Correlation coefficients among

gut microbiota, phenotypic traits, and DEGs were calculated using the "ggplot" package in R (v3.3.1), and correlation network heatmaps were constructed. The top 20 most abundant gut microbiota at the taxonomic level were selected, and co-occurrence networks were generated based on absolute correlation coefficients $|r| \geq 0.5$ and $P$-value $<0.05$. Data were visualized using the Cytoscape software.

## RESULTS

### Phenotypic trait analysis of male GFP morphotypes at various developmental stages

Tables 1 to 3 present the phenotypic traits and coefficients of variation (CV) for BCs, OCs, and SMs at 100, 110, and 120 days post-hatching, respectively. The measurements of phenotypic traits, including BL, AL, AD, CW2, and CD, revealed CV values mostly ranging between 1% and 10%, indicating stable and reliable phenotypic data. During the first 10 days of the experiment, the claw weight of BCs increased by 1.02 g, representing the fastest development among all male morphotypes, which ensures the dominance of BCs in the social hierarchy. Compared to day 100, the body weight of OCs nearly doubled by day 110, showing the highest growth rate among all morphotypes. Throughout the experimental period, SMs consistently exhibited the lowest body weight and claw weight, and therefore, they remain at the bottom of the social hierarchy (Fig. 2).

### Gut microbiota diversity analysis of male GFP morphotypes at different developmental stages

The intestine of GFPs is linear and slender (Fig. 3A), which allows rapid nutrient metabolism. This suggests a critical role of gut microbiota in the development of GFPs. To investigate the role of gut microbiota in the differentiation of male morphotypes, 16S rRNA amplicon sequencing was performed on the intestinal samples collected at key developmental stages (100, 110, and 120 days post-hatching). After quality control and filtering raw sequences from 54 samples, a total of 3,116,208 high-quality sequences were obtained. Based on the operational taxonomic units (OTUs) microbial classification, these sequences were clustered into 755 OTUs at a 97% similarity threshold, with 22–323 OTU counts per sample. The taxonomic analysis identified 25 phyla, 391 genera, and 539 species (Tables S1 and S2). The rarefaction curves plateaued, indicating sufficient sequencing depth to reliably estimate species richness.

**TABLE 1** The phenotypic traits of male GFP at the age of 100 days[a]

| Days | 100 days | | | | | |
|---|---|---|---|---|---|---|
| **Trait** | **BC** | | **OC** | | **SM** | |
| | Mean ± SD /g/mm | CV/% | Mean ± SD /g/mm | CV/% | Mean ± SD /g/mm | CV/% |
| TW/g | 31.58 ± 2.55 | 8.07% | 10.66 ± 1.42 | 13.32% | 3.09 ± 0.43 | 13.92% |
| BL/mm | 104.60 ± 3.03 | 2.90% | 75.99 ± 3.59 | 4.72% | 50.6 ± 2.3 | 4.55% |
| AL/mm | 66.96 ± 2.77 | 4.14% | 48.92 ± 2.23 | 4.56% | 33.91 ± 1.24 | 3.66% |
| AW2/mm | 15.52 ± 1.45 | 9.35% | 11.93 ± 1.21 | 10.14% | 7.19 ± 0.52 | 7.23% |
| AD/mm | 21.32 ± 1.82 | 8.54% | 14.9 ± 0.55 | 3.69% | 10.37 ± 0.54 | 5.21% |
| CL/mm | 38.91 ± 2.59 | 6.66% | 26.2 ± 1.52 | 5.80% | 16.99 ± 1.86 | 10.95% |
| CW2/mm | 23.95 ± 1.57 | 6.55% | 14.83 ± 1.13 | 7.62% | 9.1 ± 0.6 | 6.59% |
| CD/mm | 27.09 ± 2.63 | 9.71% | 17.16 ± 1.22 | 7.11% | 10.77 ± 0.86 | 7.99% |
| SPD/mm | 11.68 ± 1.28 | 10.96% | 8.72 ± 0.98 | 11.24% | 5.61 ± 0.36 | 6.42% |
| SPW/mm | 13.59 ± 1.48 | 10.89% | 9.11 ± 0.66 | 7.24% | 4.88 ± 0.29 | 5.94% |
| PL/mm | 20.72 ± 1.41 | 6.81% | 11.22 ± 1.21 | 10.78% | 5.13 ± 0.75 | 14.62% |
| PW/mm | 3.96 ± 0.33 | 8.34% | 1.96 ± 0.2 | 10.20% | 0.98 ± 0.15 | 15.31% |
| AW1/g | 11.91 ± 0.74 | 6.21% | 4.75 ± 0.78 | 16.42% | 1.39 ± 0.2 | 14.39% |
| CW1/g | 18.14 ± 2.40 | 13.23% | 4.92 ± 0.86 | 17.48% | 1.35 ± 0.24 | 17.78% |
| WSL/g | 1.71 ± 0.26 | 15.19% | 0.2 ± 0.05 | 25.00% | 0.02 ± 0.01 | 50.00% |

[a]The unit of weight is in grams (g); the unit of length, depth, and width is in millimeters (mm).

**TABLE 2** The phenotypic traits of male GFP at the age of 110 days[a]

| Days | 110 days | | | | | |
|---|---|---|---|---|---|---|
| Trait | BC | | OC | | SM | |
| | Mean ± SD /g/mm | CV/% | Mean ± SD /g/mm | CV/% | Mean ± SD /g/mm | CV/% |
| TW/g | 41.23 ± 3.1 | 7.52% | 19.43 ± 1.49 | 7.67% | 5.13 ± 0.79 | 15.40% |
| BL/mm | 112.95 ± 4.61 | 4.08% | 90.81 ± 2.47 | 2.72% | 60.55 ± 3.43 | 5.66% |
| AL/mm | 72.06 ± 2.02 | 2.80% | 60.99 ± 3.37 | 5.53% | 41.55 ± 2.17 | 5.22% |
| AW2/mm | 16.85 ± 1.32 | 7.83% | 13.84 ± 1.21 | 8.74% | 8.07 ± 0.88 | 10.90% |
| AD/mm | 22.57 ± 0.78 | 3.46% | 17.71 ± 0.53 | 2.99% | 11.61 ± 1.01 | 8.70% |
| CL/mm | 40.27 ± 4.45 | 11.05% | 32.76 ± 2.19 | 6.68% | 19.11 ± 1.14 | 5.97% |
| CW2/mm | 24.78 ± 1.14 | 4.60% | 18.94 ± 0.89 | 4.70% | 11.04 ± 0.61 | 5.53% |
| CD/mm | 29.31 ± 0.43 | 1.47% | 21.48 ± 1.03 | 4.80% | 13.2 ± 1.11 | 8.41% |
| SPD/mm | 12.71 ± 1.64 | 12.90% | 8.63 ± 1.55 | 17.96% | 6.05 ± 0.92 | 15.21% |
| SPW/mm | 14.66 ± 0.64 | 4.37% | 11.45 ± 0.64 | 5.59% | 7.5 ± 0.59 | 7.87% |
| PL/mm | 23.72 ± 1.3 | 5.48% | 15.28 ± 0.94 | 6.15% | 8 ± 0.56 | 7.00% |
| PW/mm | 4.54 ± 0.43 | 9.47% | 2.58 ± 0.45 | 17.44% | 1.42 ± 0.18 | 12.68% |
| AW1/g | 14.68 ± 1.5 | 10.22% | 7.92 ± 0.68 | 8.59% | 2.38 ± 0.39 | 16.39% |
| CW1/g | 19.58 ± 1.45 | 7.41% | 10.17 ± 1.25 | 12.29% | 2.27 ± 0.39 | 17.18% |
| WSL/g | 2.73 ± 0.59 | 21.61% | 0.57 ± 0.2 | 35.09% | 0.05 ± 0.01 | 20.00% |

[a]For the definitions of the abbreviations employed in this table, refer to the footnote of Table 1.

Alpha diversity analysis was conducted to evaluate the richness, diversity, and completeness of gut microbiota in male GFPs at the three time points. As shown in Fig. 3B, the coverage index exceeded 99.8% for all samples, which demonstrates that the sequencing results captured the majority of bacterial communities. Microbial richness, as measured by the Ace and Chao indices, was higher at 100 days post-hatching for BCs, OCs, and SMs than they were at 110 and 120 days. The Chao index revealed significant correlations in microbial richness among BC1, OC1, and SM1, as well as among SM1, BC2, OC2, SM2, and BC3. Shannon and Simpson indices indicated no significant differences in microbial diversity and richness among BCs, OCs, and SMs across the three time points. However, the Simpson index showed that SM1 at 100 days post-hatching exhibited higher microbial diversity and richness than that demonstrated by the other groups.

Venn diagram analysis of the three time points (Fig. 3C) revealed 97 shared core OTUs at 100 days post-hatching, 52 at 110 days, and 58 at 120 days. The largest difference

**TABLE 3** The phenotypic traits of male GFP at the age of 120 days[a]

| Days | 120 days | | | | | |
|---|---|---|---|---|---|---|
| Trait | BC | | OC | | SM | |
| | Mean ± SD /g/mm | CV/% | Mean ± SD /g/mm | CV/% | Mean ± SD /g/mm | CV/% |
| TW/g | 44.68 ± 4.04 | 9.04% | 25.82 ± 1.09 | 4.22% | 12.05 ± 1.7 | 14.11% |
| BL/mm | 117.82 ± 2.28 | 1.94% | 100.73 ± 2.47 | 2.45% | 79.53 ± 2.94 | 3.70% |
| AL/mm | 75.67 ± 2.00 | 2.64% | 64.59 ± 2.01 | 3.11% | 52.57 ± 1.74 | 3.31% |
| AW2/mm | 16.73 ± 0.88 | 5.26% | 14.75 ± 1.50 | 10.17% | 10.57 ± 0.29 | 2.74% |
| AD/mm | 22.21 ± 0.46 | 2.07% | 19.18 ± 0.64 | 3.34% | 14.88 ± 0.8 | 5.38% |
| CL/mm | 42.44 ± 1.21 | 2.85% | 35.99 ± 2.35 | 6.53% | 25.82 ± 1.31 | 5.07% |
| CW2/mm | 25.60 ± 1.67 | 6.52% | 20.97 ± 0.64 | 3.05% | 15.37 ± 0.91 | 5.92% |
| CD/mm | 29.42 ± 1.39 | 4.72% | 24.43 ± 0.68 | 2.78% | 17.56 ± 0.72 | 4.10% |
| SPD/mm | 12.57 ± 0.93 | 7.40% | 10.72 ± 1.40 | 13.06% | 7.77 ± 0.5 | 6.44% |
| SPW/mm | 14.83 ± 0.69 | 4.65% | 12.89 ± 0.69 | 5.35% | 9.56 ± 0.26 | 2.72% |
| PL/mm | 24.67 ± 3.31 | 13.42% | 15.88 ± 1.72 | 10.83% | 11.33 ± 0.79 | 6.97% |
| PW/mm | 4.77 ± 0.63 | 13.21% | 2.71 ± 0.33 | 12.18% | 1.94 ± 0.26 | 13.40% |
| AW1/g | 15.55 ± 1.64 | 10.55% | 10.68 ± 0.81 | 7.58% | 5.15 ± 0.69 | 13.40% |
| CW1/g | 25.87 ± 3.76 | 14.53% | 14.12 ± 0.79 | 5.59% | 6.18 ± 1.05 | 16.99% |
| WSL/g | 2.50 ± 1.27 | 50.80% | 0.95 ± 0.44 | 46.32% | 0.24 ± 0.14 | 58.33% |

[a]For the definitions of the abbreviations employed in this table, refer to the footnote of Table 1.

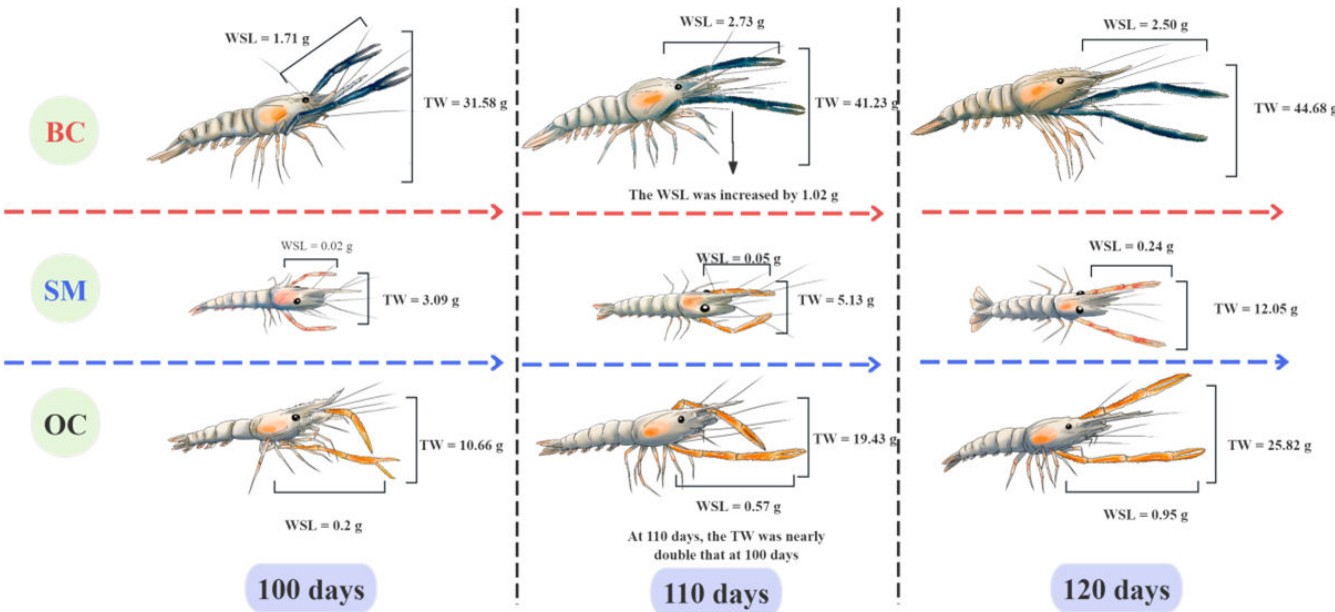

**FIG 2** Morphotype differentiation process of male giant freshwater prawn. .

in OTU counts was observed between SM1 (290) and BC1 (24) at 100 days, while the differences between SM and BC OTU counts decreased at 110 and 120 days. Each group contained multiple unique OTUs, and a total of 34 core OTUs were shared across all groups at the 3 time points. The taxonomic composition of gut microbiota at the phylum, genus, and species levels is shown in Fig. S2. At the phylum level, Firmicutes and Proteobacteria were the dominant phyla in all males. At the genus level, *Lactococcus* was the most abundant genus in nearly all samples, while at the species level, *L. garvieae* and *Lactobacillus taiwanensis* were the most abundant species. Principal coordinate analysis (PCoA) and non-metric multidimensional scaling (NMDS) revealed no significant differences among the three morphotypes. However, partial least squares-discriminant analysis (PLS-DA) showed a clear separation of SMs from BCs and OCs (Fig. 4).

Heatmaps were generated for the top 20 most abundant phyla, genera, and species to visualize abundance distribution trends and identify group differences (Fig. 5). At the phylum level, Firmicutes and Proteobacteria were the most abundant phyla across all groups (Fig. 5A). At the genus level, *Lactococcus* was the most abundant genus, while *Acinetobacter* and *Aeromonas* were more abundant in SMs than in other morphotypes. *Enterococcus* was widely distributed and highly abundant across all groups, while *Acinetobacter* and *Exiguobacterium* abundances were the highest in SM2 (Fig. 5B). At the species level, *L. garvieae*, *L. taiwanensis*, and *Enterococcus faecalis* were widely distributed, with *L. garvieae* and *L. taiwanensis* being the most abundant (Fig. 5C). Kruskal-Wallis rank-sum tests ($P < 0.05$, $P < 0.01$) were performed to assess significant differences regarding the dominant phyla, genera, and species among BCs, OCs, and SMs (Fig. 5D through F). At the phylum level, Actinobacteriota and Bacteroidota showed significant or highly significant differences among morphotypes, with higher abundances in SMs. At the genus level, 10 genera showed significant differences, among which *Achromobacter* and *Exiguobacterium* exhibited the largest differences (*Achromobacter* was more abundant in BCs, while *Exiguobacterium* was more abundant in SMs). At the species level, *L. garvieae* and *L. taiwanensis* showed significant differences among the morphotypes; *L. garvieae* was more abundant in SMs and *L. taiwanensis* in BCs.

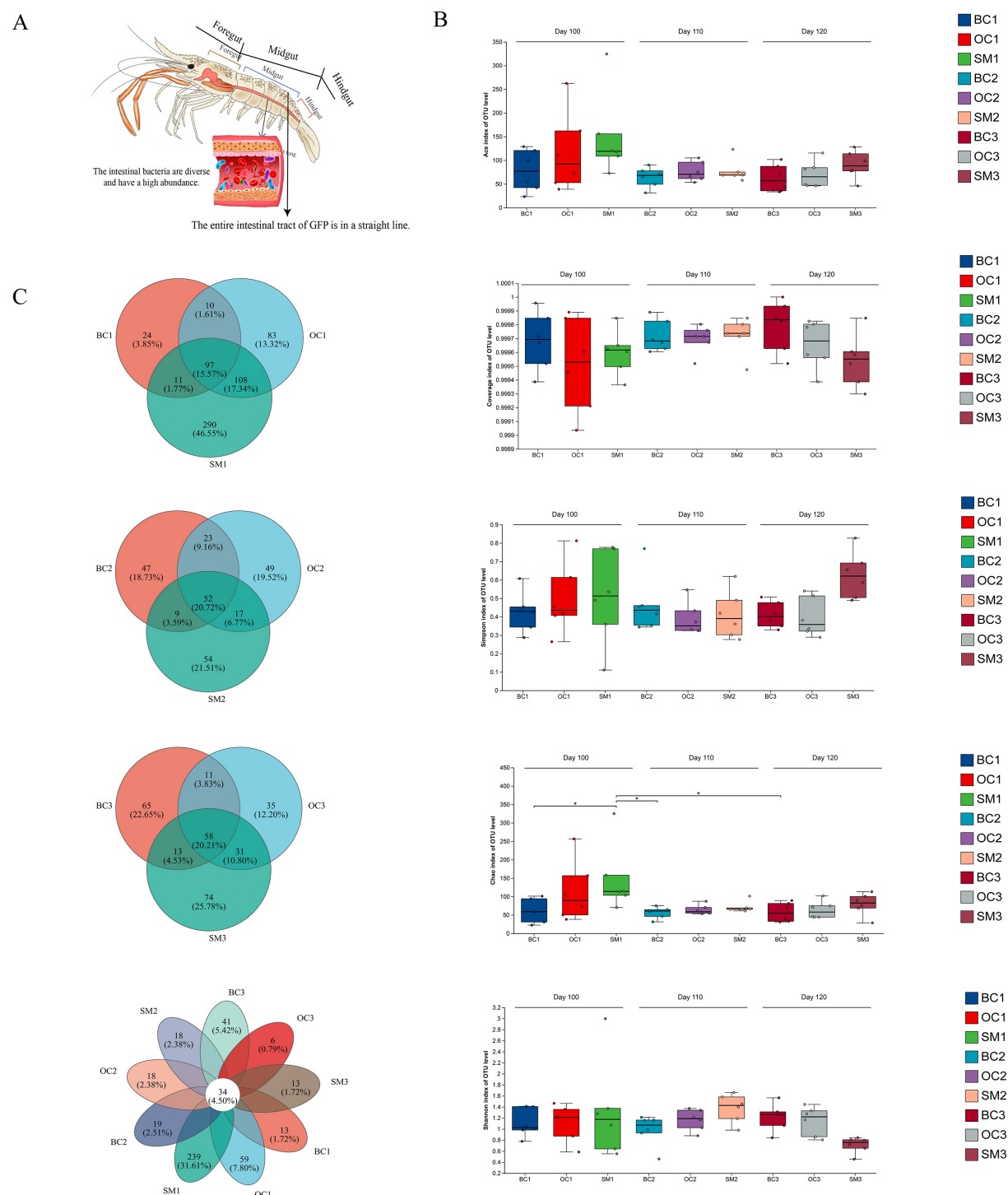

**FIG 3** Biodiversity analysis of intestinal microbiota in male giant freshwater prawn. (A) Intestine of GFP. (B) Analysis of α-diversity of BC, OC, and SM intestinal flora at 100, 110, and 120 days of age. (C) The Venn diagram of intestinal flora species composition of BC, OC, and SM at 100, 110, and 120 days of age.

## Gonadal transcriptome analysis of male GFP morphotypes

To elucidate the regulatory mechanisms of host genes in male GFP morphotype differentiation, transcriptome analysis was performed on 54 gonadal samples. A total of 364.67 GB of clean data were obtained, with each sample yielding over 6 GB of clean data and a Q30 base percentage of >89.65%. Quality control metrics for the raw data

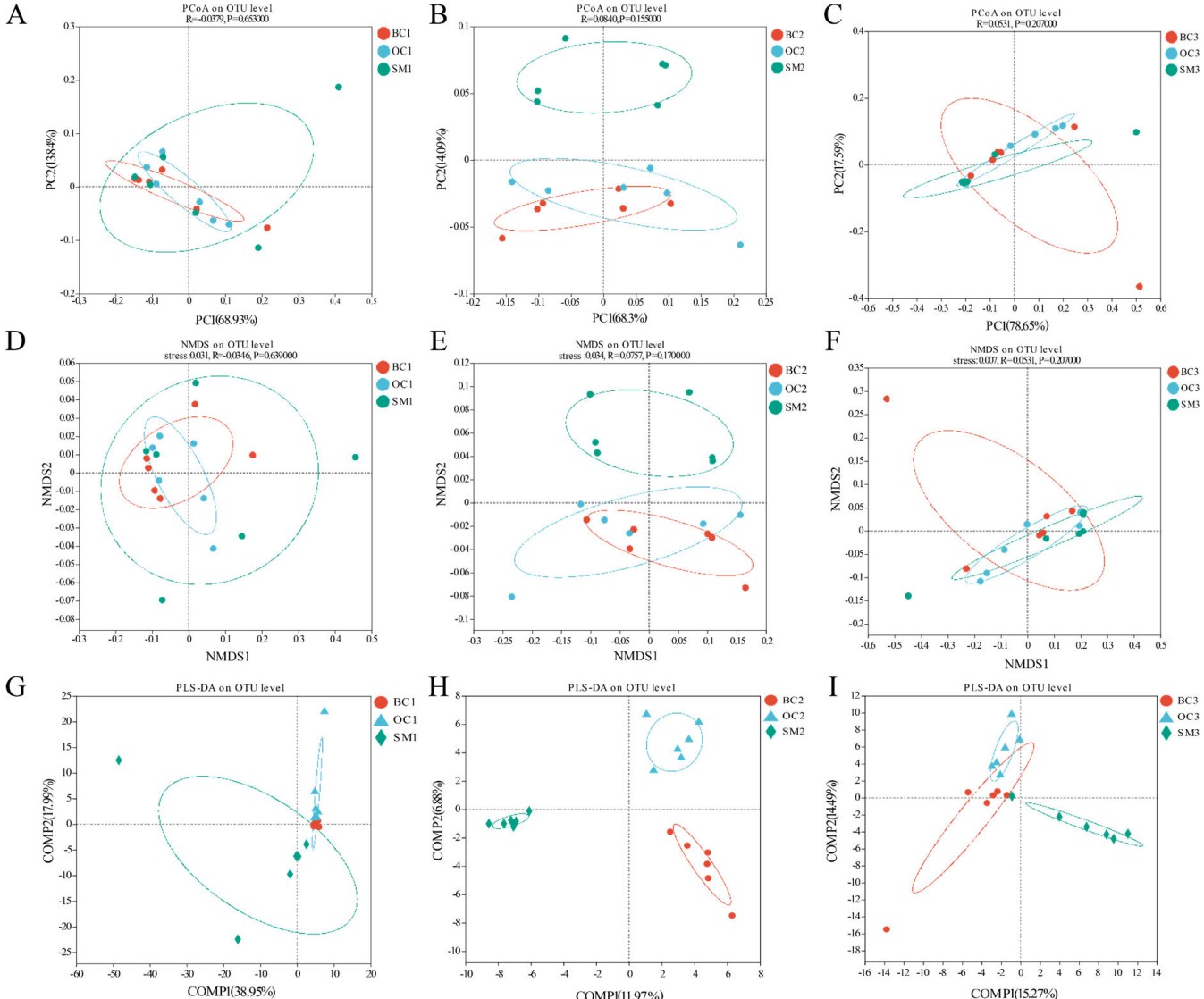

**FIG 4** PCoA, NMDS, and PLS-DA analyses of intestinal flora of giant freshwater prawn. (A–C) PCoA of the intestinal flora of BC, OC, and SM groups at 100, 110, and 120 days of age. (D–F) NMDS of the intestinal flora of BC, OC, and SM groups at 100, 110, and 120 days of age. (G–I) PLS-DA of the intestinal flora of BC, OC, and SM groups at 100, 110, and 120 days of age.

are summarized in Table S3. The clean reads ranged from 40,301,430 to 61,850,906 per sample, with average Q20 and Q30 ratios of 96.94% and 91.41%, respectively. The average error rate was 0.01%, and the average GC content was 44.3%. These results indicate high sequencing quality, which ensures the accuracy of the subsequent analyses.

*De novo* assembly of the clean data generated a total of 172,383 transcripts and 99,565 unigenes (Table S4). The transcripts comprised 232,050,629 bases, with the largest transcript being 33,376 bp, the smallest being 201 bp, and an average length of 1,346.13 bp. The N50 length was 2,646 bp, the E90N50 length was 4,802 bp, and the GC content was 40.38%. For unigenes, the total base count was 131,330,804, with the largest unigene being 33,376 bp, the smallest 201 bp, and an average length of 1,319.05 bp. The N50 length was 2,504 bp, the E90N50 length was 5,406 bp, and the GC content was 40.05%.

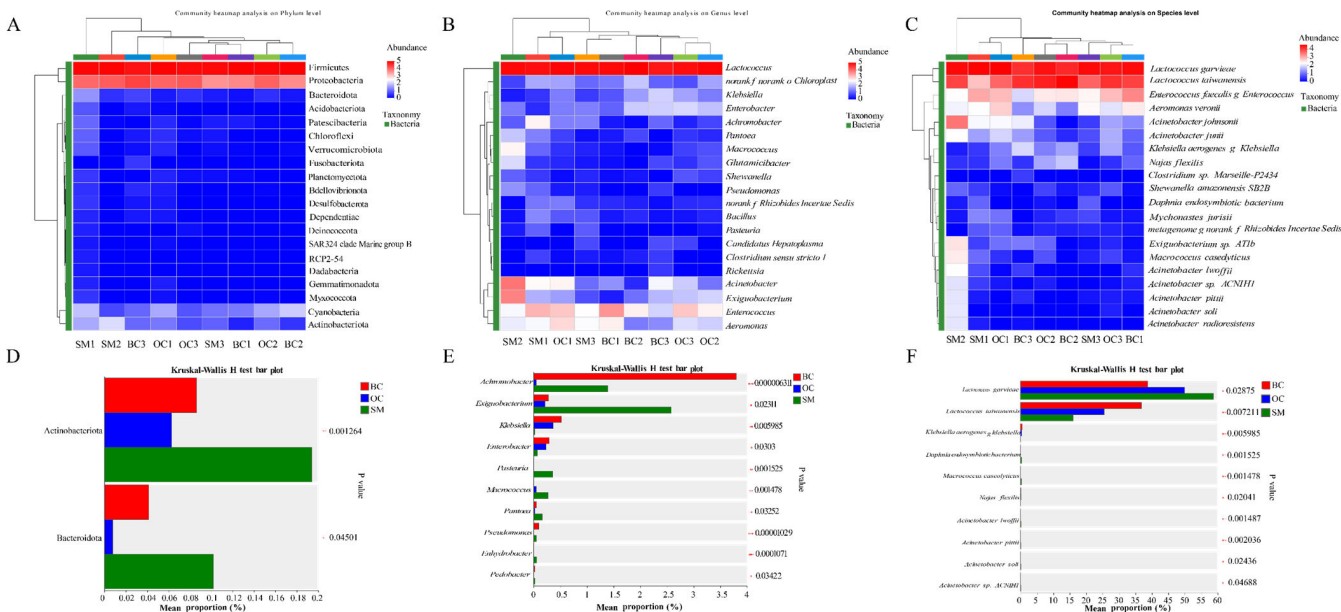

**FIG 5** Heatmap of the intestinal flora of giant freshwater prawn males and the analysis of the significance of differences among them. (A) and (D) Phylum-level intestinal flora composition heatmap and inter-group difference significance test analysis. (B) and (E) Generic-level intestinal flora composition heatmap and inter-group difference significance test analysis. (C) and (F) Species-level intestinal flora composition heatmap and inter-group difference significance test analysis.

## Functional annotation and classification of unigenes in male GFP morphotypes

As shown in Fig. S3A, a total of 99,541 genes were annotated. Of these, 17,059 (17.14%), 10,834 (10.88%), 18,506 (18.59%), 26,167 (26.29%), 13,089 (13.15%), and 16,337 (16.41%) genes were annotated to the GO, KEGG, EggNOG, NR, Swiss-Prot, and Pfam databases, respectively. Using an $E$-value threshold of 1E−5, the NR database revealed that the annotated unigenes showed high homology with the genes of various shrimp and crab species. Two species with the highest number of homologous sequences were *Homarus americanus* (15.35%) and *Penaeus vannamei* (11.21%) (Fig. S3B). In the GO database, 17,059 unigenes were annotated to 3 main functional categories: biological process (BP), cellular component (CC), and molecular function (MF) (Fig. S3C). Within the BP category, "cellular process" and "metabolic process" contained the highest number of unigenes (7,826 and 7,606, respectively). In the CC category, "cell part," "membrane part," and "protein-containing complex" were significantly enriched with 5,604, 3,917, and 2,696 unigenes, respectively. In the MF category, "binding" and "catalytic activity" had the highest number of unigenes, with 10,419 and 7,760, respectively. Additionally, 10,834 unigenes annotated in the KEGG database were mapped to 6 specific pathways: environmental information processing, human diseases, cellular processes, organismal systems, genetic information processing, and metabolism (Fig. S3D). The pathway with the highest number of unigenes was "Signal transduction" (1,281), followed by "Infectious disease: viral" (1,150) and "Cancer: overview" (1,024).

## Differential gene expression analysis in male GFP morphotypes

To explore the molecular mechanisms underlying male GFP morphotype differentiation, we compared gene expression levels among BCs, OCs, and SMs at 100, 110, and 120 days post-hatching. A total of 10,523 DEGs were identified (Fig. S3E). DEGs were defined using thresholds of |fold-change| >2 and $P$-adjust <0.05. The volcano plots of DEGs at 100, 110, and 120 days are presented in Fig. S4. At 100 days, the comparison between OC1 and SM1 yielded the maximum DEGs (1,313; 906 upregulated and 407 downregulated). At

110 days, a comparison between BC2 and OC2 yielded the highest number of DEGs (416; 85 upregulated and 331 downregulated). At 120 days, only one upregulated DEG was identified between BC3 and SM3. The number of DEGs between BCs and SMs decreased over time, while that between BC and OC increased. The number of DEGs between OCs and SMs initially decreased and then increased. A comparison of DEGs within the same morphotype at different developmental stages is shown in Fig. S5. For BCs, 1,820 DEGs were identified between BC1 and BC2, 1,175 between BC1 and BC3, and 1,009 between BC2 and BC3. For OCs, 1,062 DEGs were identified between OC1 and OC2, 2,883 between OC1 and OC3, and 1,977 between OC2 and OC3. For SMs, 880 DEGs were identified between SM1 and SM2, 1,920 between SM1 and SM3, and 872 between SM2 and SM3. Overall, fewer DEGs were identified during the mid-to-late stages of morphotype differentiation (110 vs 120 days) compared to the numbers in the early-to-late stages (100 vs 120 days).

## Functional annotation of DEGs using GO and KEGG

KEGG and GO annotations were performed to further elucidate the functional roles of DEGs in male GFP morphotypes at 100, 110, and 120 days post-hatching (Fig. S6). Except for 100 days (no DEGs were annotated to genetic information processing), DEGs at 110 and 120 days were annotated to six KEGG categories: metabolism, genetic information processing, environmental information processing, cellular processes, organismal systems, and human diseases. Additionally, DEGs across all three time points were annotated to KEGG pathways related to growth, energy metabolism, immunity, and disease, including Infectious disease: bacterial, Cancer: specific types, Cancer: overview, Infectious disease: viral, Cardiovascular disease, Digestive system, Immune system, Cellular community-eukaryotes, Signal transduction, and Carbohydrate metabolism. GO annotation revealed that, except for the 120-day group (the protein-containing complex in the CC category had the highest number of DEGs), the BP categories (cellular and metabolic processes), the CC categories (cell and membrane parts), and the MF categories (catalytic activity and binding) were consistently annotated across all time points, with higher numbers of DEGs than those in other pathways.

## KEGG enrichment analysis of DEGs

KEGG enrichment analysis was performed on the corrected DEG sets using a threshold of $P$-adjust <0.05. Figure S7 shows the top 20 most significantly enriched KEGG pathways for BC vs OC, BC vs SM, and OC vs SM at 100, 110, and 120 days. At 100 days, BC1 vs SM1, OC1 vs SM1, and BC1 vs OC1 were all enriched in pancreatic secretion. At 110 days, BC2 vs SM2, OC2 vs SM2, and BC2 vs OC2 were enriched in dilated cardiomyopathy and cardiac muscle contraction. At 120 days, BC3 vs SM3 was enriched only in lysine degradation, while OC3 vs SM3 and BC3 vs OC3 were enriched in ribosome, Alzheimer's disease, oxidative phosphorylation, pathways of neurodegeneration-multiple diseases, thermogenesis, and retrograde endocannabinoid signaling. Figure S8 shows the top 20 most significantly enriched KEGG pathways for DEGs within the same morphotype at different developmental stages. The BC1 vs BC2, BC1 vs BC3, and BC2 vs BC3 were all enriched in dilated cardiomyopathy; the OC1 vs OC2, OC1 vs OC3, and OC2 vs OC3 were all enriched in diabetic cardiomyopathy. No common KEGG pathways were enriched in SM1 vs SM2, SM1 vs SM3, or SM2 vs SM3. Several KEGG signaling pathways associated with male GFP morphotype differentiation were identified, including the Hippo signaling pathway, MAPK signaling pathway, Wnt signaling pathway, apoptosis, and calcium signaling pathway. Through KEGG pathway analysis, key DEGs influencing male GFP morphotype differentiation were identified, including protein *Wnt-6*, cathepsin B (*CTSB*), cathepsin L (*CTSL*), cellular tumor antigen p53 (*TP53*), fatty acid synthase (*FAS*), heat shock protein 70 (Hsp70), serum amyloid A (*SAA*), differentially expressed in FDCP 6 (*DEF6*), calpain clp-1, and prophenoloxidase activating enzyme (*PPAE*).

## Association analysis of phenotypic traits, gut microbiota, and DEGs

Figure 6A and B present the heatmap and network diagram illustrating the associations between phenotypic traits and gut microbiota. *L. taiwanensis* showed significant or highly significant positive correlations with 15 phenotypic traits, while *L. garvieae*, *Acinetobacter johnsonii*, and *Macrococcus caseolyticus* exhibited significant or highly significant negative correlations with these traits. Specifically, *L. taiwanensis* was significantly positively correlated with PL, BL, CW1, AD, TW, PW, and AW2. In contrast, *Acinetobacter radioresistens* showed significant negative correlations with all 15 phenotypic traits. At the genus level, TRINITY_DN275_c0_g1 (*PPAE*) exhibited significant correlations with numerous bacterial genera (Fig. 6C). *Enterococcus* was significantly or highly significantly positively correlated with TRINITY_DN18337_c0_g1 (juvenile hormone esterase, *JHE*), TRINITY_DN11174_c0_g1 (CLIP-LEC), TRINITY_DN17821_c0_g1 (balbani ring protein 3), TRINITY_DN3047_c1_g1 (procollagen-lysine,2-oxoglutarate 5-dioxygenase 1, *PLOD1*), TRINITY_DN93139_c0_g2 (Calpain clp-1), and TRINITY_DN12183_c1_g1 (bestrophin-4, *BEST4*). At the species level, *Mychonastes jurisii* showed highly significant positive correlations with TRINITY_DN79496_c1_g1 (*TP53*), TRINITY_DN11690_c0_g3 (trypsin), TRINITY_DN11887_c0_g2 (*CTSB*), and TRINITY_DN31284_c0_g1 (*CTSL*) (Fig. 6D). TRINITY_DN186216_c0_g1 (voltage-dependent anion-selective channel, *VDAC*) exhibited highly significant negative correlations with *A. radioresistens* and *Acinetobacter soli*. Additionally, *PPAE*, calpain clp-1, and *BEST4* showed significant or highly significant negative correlations with *A. radioresistens*, *Exiguobacterium* sp. AT1b, and *A. soli*. Figure 6E shows the association analysis between DEGs and phenotypic traits. TRINITY_DN2197_c0_g3 (single insulin-binding domain protein, *SIBD*) was significantly positively correlated with all 15 phenotypic traits, while trypsin was highly significantly negatively correlated with these traits. The *CTSL* showed significant or highly significant negative correlations with all 15 traits. *CTSB* was significantly or highly significantly negatively correlated with AL, AD, CW2, SPW, TW, AW1, BL, CW1, SPD, and AW2. *PPAE* and calpain clp-1 were significantly positively correlated with SPD, while *VDAC* showed significant positive correlations with AL and PL.

To further investigate the relationship between morphotype differentiation-related genes and phenotypic traits during male GFP development, a co-expression gene network was constructed using weighted gene co-expression network analysis (WGCNA). In total, 17 co-expression modules were identified, with the number of genes per module ranging from 71 to 12,214 (Fig. 6F). The MEturquoise module contained the maximum genes (12,214), while the MElightcyan module had the fewest (46). The MEyellow module showed the highest and most significant positive correlations with TW, AL, BL, CL, SPW, and PL. In contrast, the MEcyan module exhibited the strongest negative correlations with TW, BL, AL, AD, CW2, CD, SPW, AW1, CW1, and WSL.

The results of the transcription, phenotype, and intestinal flora association network analysis indicated that *Achromobacter* was linked to 15 phenotypes (Fig. 6G). Additionally, TRINITY_DN181316_c1_g1 (putative histone H3.3) was significantly associated with several transcripts and *Exiguobacterium*. Calpain clp-1 was associated with PL, CD, and BL, and *BEST4* was linked to CL and PL. *Enterobacter* was associated with the PL and SPD phenotypes. TRINITY_DN174412_c0_g1 (*Hsp3*) was connected to the transcripts of putative histone H3.3. Analyzed as a whole, the results showed that associative model "*Achromobacter*—phenotypes—TRINITY_DN93139_c0_g2 (calpain clp-1)" might play an important role in the differentiation process of GFP male castes.

## DISCUSSION

Extensive research has demonstrated that the morphotype differentiation of male GFPs is a result of the combined effects of intrinsic and extrinsic factors. Extrinsic factors include environmental conditions, social hierarchy based on individual size, and nutritional differences (47), while intrinsic factors encompass genetic variation, age, gene expression, and gut microbiota regulation (29, 48, 49). Previous studies

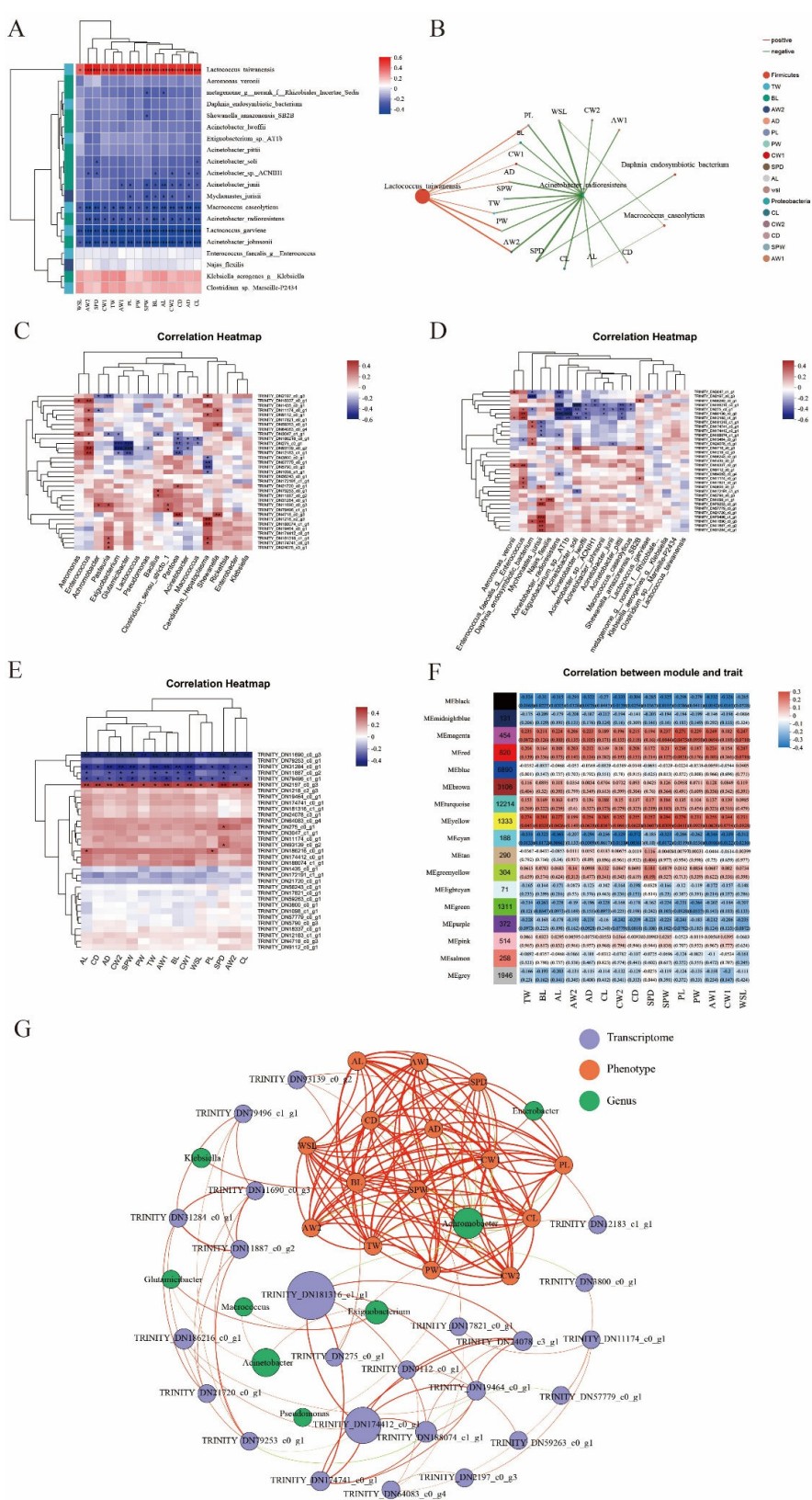

**FIG 6** Correlation analysis among intestinal flora, phenotypic traits, and host genes. (A) and (B) Heatmap and network map of association analysis between intestinal flora and phenotypic traits. (C) and (D) Heatmap of association analysis between DEGs and intestinal flora (genus and species level). (E) Heatmap of correlation analysis between phenotypic traits and DEGs. (Continued on next page)

Fig 6 (Continued)

(F) Correlation analysis between weighted gene co-expression network analysis gene modules and phenotypic traits (abscissa is phenotypic traits, and ordinate is the name of each gene module; the column of numbers on the left indicates the number of genes/transcripts of the module, and each set of data on the right indicates the correlation coefficient and significance *P*-value of the module with the phenotype). (G) Analysis of intestinal flora, phenotypic traits, and host gene association network.

have identified potential DEGs, gut microbiota, and metabolites influencing male GFP morphotype differentiation (13, 15, 29, 50). However, a systematic analysis of the developmental patterns and underlying mechanisms of male GFP morphotype differentiation remains lacking. To address this gap, our study is the first to sample male GFPs at key developmental time points (100, 110, and 120 days post-hatching) and perform multi-omics analyses. By integrating multi-omics data, we revealed the dynamic changes in phenotypic traits during the development of male GFPs and identified key gut microbiota and potential functional genes. Furthermore, association analysis was employed to construct correlation networks, which provided valuable insights into the molecular mechanisms underlying male GFP morphotype differentiation.

Despite the significant differences in claw color and spine morphology between SMs, BCs, and OCs (16, 36), the statistical analysis of the phenotypic data obtained from BCs, OCs, and SMs at the aforementioned three developmental stages revealed that BCs exhibited rapid growth of the second leg (claw) during the first 10 days of morphotype differentiation. As the second leg serves as a powerful weapon for competition over food and space among GFPs, its rapid development likely enables the BC to secure a dominant position in the social hierarchy, ensuring its status at the top of the social ladder. Although the OC had a significantly lower weight of body and second leg compared to those of the BC, its high growth rate suggests that it has a considerable growth potential (51). Notably, the OC nearly doubled its body weight during the first 10 days of morphotype differentiation, making it a key male phenotype for GFP breeding. In contrast, the SM consistently exhibited significantly lower weight of body and second leg compared to those in BC and OC, which may confer greater mobility, allowing the SM to escape intense social conflicts within the population. Based on these findings, male GFPs can be categorized into fast-growing morphotypes (BC and OC) and slow-growing morphotypes (SM).

Gut microbiota, often referred to as "phenotype and behavior modulators" of the host (23), play a crucial role in host development (52) and are closely associated with male GFP morphotype differentiation (29). An analysis of the gut microbiota structure of the three male GFP morphotypes revealed Firmicutes and Proteobacteria as the dominant phyla and *Lactococcus* as the most abundant genus. These results are consistent with those of previous studies (28, 29). However, as the males aged, the abundance of Firmicutes decreased in BCs and OCs, while that of Proteobacteria increased. In contrast, SMs showed the opposite trend in Firmicutes and Proteobacteria abundance. Given that many Proteobacteria members are known to enhance protein and carbohydrate metabolism (53, 54), we hypothesize that a higher Proteobacteria abundance may promote energy metabolism in the GFP gut, thereby facilitating the development of fast-growing morphotypes (BC and OC). Additionally, *Lactococcus* was highly abundant in all three morphotypes throughout development. As a native probiotic, *Lactococcus* is more likely to colonize the intestines of BCs, OCs, and SMs (55) and significantly improves the growth performance (56), influencing male GFP morphotype differentiation. Notably, *L. garvieae* abundance was consistently higher in SM than it was in BC and OC, while *L. taiwanensis* abundance was higher in BC compared to that in OC and SM. Previous studies have identified *L. garvieae* as a pathogen that severely impacts the survival rates of various farmed fish species (57, 58). Therefore, the persistent high abundance of *L. garvieae* in SM suggests it to be a potential factor that hinders GFP development. In contrast, the high abundance of *L. taiwanensis* in BC and OC may promote growth, as this

species is capable of hydrolyzing plant polysaccharides (59), which have been shown to enhance immune function, reduce inflammation, and improve the overall physiological health (60). Thus, we hypothesize that *L. taiwanensis* may support GFP growth and development by strengthening immune function and optimizing the gut microbiota structure.

Significant differences in the abundance of *Achromobacter* and *Exiguobacterium* were identified through an inter-group differential analysis among the three morpho-types. *Achromobacter* is known to produce secondary metabolites with antimicrobial and antifungal properties (61), while members of *Exiguobacterium* can utilize various complex polysaccharides and proteins, enabling its survival in diverse environments (62). These two bacterial genera were particularly abundant in BC, which exhibits stronger disease resistance compared to OC. Therefore, *Achromobacter* and *Exiguobacterium* may contribute to enhancing the disease resistance of GFP. Through association analysis between gut microbiota and phenotypic traits, we observed significant differences in the abundance of *L. taiwanensis* and *L. garvieae* between BC and SM morphotypes. Heatmap analysis revealed that *L. taiwanensis* showed significant or highly significant positive correlations with all 15 phenotypic traits, whereas *L. garvieae* exhibited significant or highly significant negative correlations with these traits. These findings suggest that *L. taiwanensis* and *L. garvieae* might play a regulatory role in GFP phenotypic traits. Furthermore, they may serve as key bacterial species influencing the differentiation of male GFP morphotypes.

Morphotype differentiation in animals is a complex process regulated by gene expression and control mechanisms, particularly in the gonads, which influence morphological development (63). In male GFPs, morphotype differences are primarily reflected in body size, which is determined by a highly coordinated and intricate process involving a precise regulation of cell number and/or cell size (64). In this study, we identified key pathways known to influence morphotype differentiation and identified multiple DEGs. Additionally, we screened for DEGs with high occurrence across all groups and conducted association analyses between these DEGs, phenotypic traits, and gut microbiota. The most representative KEGG pathways identified in this study include the Hippo signaling pathway, apoptosis, and Wnt signaling pathway. The Hippo signaling pathway regulates critical physiological processes, such as cell proliferation, differentiation, embryogenesis, and tissue regeneration (65). Apoptosis is essential for normal cell turnover, immune system development, and programmed cell death (66). These pathways may influence cell renewal and immune system regulation, thereby affecting body size and immune function in male GFPs and contributing to morphotype differentiation. Furthermore, the Wnt signaling pathway has been shown to promote organ and tissue regeneration in various organisms (67). In zebrafish, Wnt/β-catenin signaling indirectly regulates cell proliferation and differentiation during regeneration, with fibroblast growth factor and Wnts synergistically promoting limb regeneration (68). Notably, *Wnt-6* plays a critical role in blastema formation during limb regeneration in *Cynops orientalis* (69). In male GFPs, claw loss or shedding often occurs during fights or molting; some OC individuals regenerate blue claws during this process. As a member of the Wnt signaling pathway, *Wnt-6* is involved in embryonic development, cell proliferation, and adult tissue homeostasis (70). Therefore, we hypothesize that *Wnt-6* activates the Wnt/β-catenin signaling pathway to regulate cell proliferation and differentiation, promoting claw and carapace regeneration and morphological changes, thereby influencing male GFP morphotype differentiation.

*CTSB* and *CTSL*, members of the lysosomal cysteine protease family, play critical roles in various biological processes, including protein degradation, apoptosis, immune responses, and tissue remodeling (71). In this study, they were significantly downregu-lated in BC vs OC, OC vs SM, and BC vs SM comparisons across all three developmen-tal stages, indicating their important roles in male GFP morphotype differentiation. Notably, *CTSB* and *CTSL* are known to regulate growth and immunity. Previous studies have demonstrated their involvement in the growth, development, and molting of

*Marsupenaeus japonicus* and *Litopenaeus vannamei* (46, 72–74). Newly molted crustaceans are more susceptible to viral infections (75), which may explain why crustaceans enhance their immune responses post-molting to improve resistance (76). We observed that *CTSB* and *CTSL* expression levels were higher in BC and OC morphotypes than they were in SM across different developmental stages. This may be attributed to the superior disease resistance and growth performance of BC and OC morphotypes relative to SM. Given the regulatory roles of *CTSB* and *CTSL* during molting, we hypothesize that these proteases play a crucial role in the development of fast-growing morphotypes (BC and OC) in GFPs.

In this study, *TP53* was significantly downregulated in BC1 vs SM1. Previous studies have reported that *TP53* can induce gut microbiota dysbiosis and increase intestinal inflammation (77). Given that BC morphotypes exhibit higher disease resistance and stronger immune responses to pathogens compared to SM, *TP53* likely plays a critical role in GFP immunity. This suggests that differences in disease resistance and immune function may also be important factors influencing morphotype differentiation. Additionally, Hsp70, which can be induced not only by heat shock but also by various stressors (78), provides protective effects against subsequent stress after initial exposure to non-lethal mild stress (79). Hsp70 also inhibits hemocyte apoptosis, ultimately enhancing stress tolerance in aquatic animals (80). In this study, Hsp70 was significantly upregulated in SM1 vs SM3 and BC2 vs BC3, indicating that stress tolerance in male GFPs improves with age across all morphotypes.

Through WGCNA of phenotypic traits and DEGs in male GFP morphotypes, this study identified key modules significantly associated with different phenotypic traits. These modules, enriched with a large number of genes, provide valuable insights into the correlations between genes and phenotypes. Furthermore, by selecting genes from the module with the highest correlation coefficients, we established a foundation for subsequent KEGG pathway analysis. Through module analysis and KEGG pathway enrichment, we identified pathways associated with morphotype differentiation and pinpointed relevant DEGs. Notably, the MEcyan module showed significant correlations with multiple phenotypic traits. We hypothesize that 188 DEGs enriched in the MEcyan module will facilitate the further screening and validation of key DEGs involved in GFP morphotype differentiation (37, 81).

Animal hosts construct their microbiomes by selecting specific microbial communities through host gene-mediated genetic effects (23, 82). Furthermore, host genes and gut microbiota collaboratively regulate phenotypic traits in animals. Through an association analysis of DEGs with phenotypic traits and gut microbiota, we identified several genes, including TRINITY_DN2197_c0_g3 (single insulin-binding domain protein), *CTSL*, *PPAE*, and calpain clp-1, significantly associated with phenotypic development. Additionally, *PPAE* and calpain clp-1 showed significant correlations with *Enterococcus*. The prophenoloxidase activating system (proPO system) plays a crucial role in the defense mechanisms of arthropods (83). The proPO system consists of prophenoloxidase, which is converted to phenoloxidase (PO) through proteolytic activation by the *PPAE* (84, 85). The PO leads to the synthesis of melanin (86), which is not only essential for immune responses (87) but also participates in physiological processes, such as wound healing and cuticle hardening (83, 86). In this study, *PPAE* was significantly upregulated in BC1 vs BC3 and SM1 vs SM2. Given the critical roles of melanin, we hypothesize that as GFPs age, the BC and SM morphotypes may exhibit enhanced immune capabilities, wound healing, and cuticle hardening due to melanin activity. These improvements likely aid BC morphotypes in territorial competition and help SM morphotypes evade attacks from BC and OC morphotypes. Furthermore, *PPAE* showed a significant positive correlation with *Enterococcus*. Previous studies have reported that *Enterococcus* is associated with improved growth performance and gut microbiota enrichment in aquatic animals (88). We hypothesize that *Enterococcus* may regulate morphological development in male GFP morphotypes. Given the strong association between *PPAE* and *Enterococcus*, we further propose that they may synergistically

influence male GFP morphotype differentiation. Finally, through the association network analysis of "intestinal bacteria—phenotypic traits—DEG," we constructed a relationship model for "*Achromobacter*—CD—TRINITY_DN93139_c0_g2 (calpain clp-1)." Calpain is an important regulatory protease that facilitates $Ca^{2+}$ activation and promotes cellular remodeling (89). The calpain gene clp-1 is associated with muscle degeneration and may generally enhance the renewal of myofibrillar proteins, helping to maintain the organized arrangement of adhesion complexes in response to sarcomere changes due to growth or cellular damage (89, 90). *Achromobacter* has been shown to be associated with animal growth and development (91). Therefore, it can be speculated that the calpain clp-1 might influence muscle growth during the development of male GFPs under the combined influence of *Achromobacter*, thereby impacting the phenotypic differentiation in male GFPs. The construction of this correlation model will enhance our understanding of the mechanisms underlying male classification in GFPs.

This study provides the first evidence in decapod crustaceans that male phenotypic differentiation arises from dynamic interactions between host genomic variation, gut microbiome assembly, and trait development. The application of tripartite association modeling (intestinal bacteria—phenotypic traits—DEG) in a non-model crustacean represents a significant technical advance. Such host-microbiome coevolutionary dynamics may generalize to other polymorphic species.

## Conclusion

In view of the male morphotype differentiation in GFP males, we carried out phenotypic measurements at the critical time points of male grading and analyzed the developmental characteristics of male morphotype differentiation. Subsequently, through 16S rRNA and gonadal transcriptomic analyses of BC, OC, and SM at different developmental stages, we identified key gut microbiota, DEGs, and development-related pathways influencing male GFP morphotype differentiation. Finally, a relationship model integrating phenotypic traits, DEGs, and gut microbiota was constructed. This study provides valuable insights into the molecular mechanisms underlying male GFP morphotype differentiation and identifies potential probiotics and key candidate genes for future breeding and genetic improvement of GFPs. We believe the study results may support the sustainable development of the GFP aquaculture industry.

## ACKNOWLEDGMENTS

We thank Elixigen Company (Shanghai, China) for their assistance in editing the manuscript. Thanks to Jiangsu Shufeng Prawn Breeding Co. Ltd. for the help with this experiment.

This work was supported by grants from the National Natural Science Foundation of China [grant number 32273121] and the earmarked fund for the China Agriculture Research System of MOF and MARA [grant number CARS-48].

Xiuxin Zhao: Writing - review & editing, Writing - original draft, Data curation. Miuying Cai: Methodology, Investigation. Shunkai Yin: Investigation, Software. Ziqi Zhou: Formal analysis, Investigation. Jie Yang: Data curation, Software. Yuqing Shen: Investigation, Software. Zhenglong Xia: Methodology. Qiongying Tang: Writing - review & editing, Supervision. Guoliang Yang: Supervision, Resources, Conceptualization. Shaokui Yi: Writing - review & editing, Methodology, Funding acquisition. Quanxin Gao: Writing - review & editing, Writing - original draft, Visualization, Validation, Methodology, Conceptualization, Funding acquisition.

The authors assert that they do not have any competing financial interests or personal relationships that could have potentially influenced the work reported in this paper.

## AUTHOR AFFILIATIONS

[1]College of Life Science, Huzhou University, Huzhou, PR China

[2]Jiangsu Shufeng Prawn Breeding Co., Ltd, Gaoyou, PR China

## AUTHOR ORCIDs

Xiuxin Zhao http://orcid.org/0009-0002-3928-9414
Shaokui Yi http://orcid.org/0000-0003-2325-0179
Quanxin Gao http://orcid.org/0000-0002-3759-8080

## FUNDING

| Funder | Grant(s) | Author(s) |
|---|---|---|
| National Natural Science Foundation of China | 32273121 | Quanxin Gao |
| Earmarked Fund for China Agriculture Research System | CARS-48 | Shaokui Yi |

## AUTHOR CONTRIBUTIONS

Xiuxin Zhao, Data curation, Writing – original draft, Writing – review and editing | Miuying Cai, Investigation, Methodology | Shunkai Yin, Investigation, Software | Ziqi Zhou, Formal analysis, Investigation | Jie Yang, Data curation, Software | Yuqing Shen, Investigation, Software | Zhenglong Xia, Methodology | Qiongying Tang, Supervision, Writing – review and editing | Guoliang Yang, Conceptualization, Resources, Supervision | Shaokui Yi, Funding acquisition, Methodology, Writing – review and editing | Quanxin Gao, Conceptualization, Funding acquisition, Methodology, Supervision, Validation, Visualization, Writing – original draft, Writing – review and editing

## DATA AVAILABILITY

The raw sequence data by the Illumina sequencing platform (SRR33015805-SRR33015858) have been deposited in the NCBI Sequence Read Archive (SRA) database under BioProject number PRJNA1247755. RNA-seq data (SRR33048285-SRR33048338) have been uploaded to the NCBI SRA database under BioProject number PRJNA1248510.

## ETHICS APPROVAL

All experimental procedures were carried out in accordance with the Chinese Guideline for Laboratory Animal Care and Use approved by the Animal Ethics Review Committee of Huzhou University (No. HUZU-20230503, Huzhou, China).

## ADDITIONAL FILES

The following material is available online.

### Supplemental Material

**Supplemental Material (Spectrum01290-25-s0001.docx).** Figures S1 to S8; Tables S1 to S4.

### Open Peer Review

**PEER REVIEW HISTORY (review-history.pdf).** An accounting of the reviewer comments and feedback.

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
