## [Reviewer comments · Microbiology Spectrum]

Microbiology Spectrum

Interaction of host gene–gut microbiota in male grading of *Macrobrachium rosenbergii*

Xiuxin Zhao, Miuying Cai, Shunkai Yin, Ziqi Zhou, Jie Yang, Yuqing Shen, Zhenglong Xia, Qiongying Tang, Guoliang Yang, Shaokui Yi, and Quanxin Gao

Corresponding Author(s): Quanxin Gao, Huzhou University

Review Timeline:

Submission Date:	April 25, 2025
Editorial Decision:	June 2, 2025
Revision Received:	July 9, 2025
Accepted:	August 1, 2025

Editor: John Chaston

Reviewer(s): Disclosure of reviewer identity is with reference to reviewer comments included in decision letter(s). The following individuals involved in review of your submission have agreed to reveal their identity: Shimming Peng (Reviewer #2); Chenghui Wang (Reviewer #3)

Transaction Report:

DOI: <https://doi.org/10.1128/spectrum.01290-25>

Re: Spectrum01290-25 (Interaction of host gene-gut microbiota in male grading of *Macrobrachium rosenbergii*)

Dear Prof. Quanxin Gao:

Thank you for the privilege of reviewing your work. Below you will find my comments, instructions from the Spectrum editorial office, and the reviewer comments.

You will see that the reviewers valued your work, although they were split on whether the manuscript was sufficiently concise. Each identified areas that, if addressed, would strengthen the contribution of the manuscript. Therefore, my decision is 'modifications', and we would be happy to consider a revised version of your manuscript. If you elect to revise the manuscript, I will send it out for another round of review. Thank you for considering Spectrum and its readership as a venue for this nice story.

Revision Guidelines

Sincerely,
John Chaston
Editor
Microbiology Spectrum

Reviewer #1 (Comments for the Author):

This study identified host KEGG pathways and key genes associated with morphotype differentiation, and further conducted association analyses to elucidate the tripartite relationships among gut microbiota, phenotypic traits, and gene expression. The

results revealed the molecular mechanisms underlying morphotype-specific phenotypic variation, providing critical insights for developmental studies and genetic improvement strategies aimed at optimizing the sustainability of GFP aquaculture. The findings of this study are novel and interesting, providing a new perspective for understanding the mechanism underlying male classification in GFPs. However, there are some issues that the authors still need to address before the paper is acceptable for publication. The detail information is as following:

Major points:

1. In Introduction, the authors state that male *Macrobrachium rosenbergii* is divided into three morphotypes (SM, OC and BC), but some studies have indicated that it can be divided into four morphotypes (SM, OC, BC and OBC). Please provide an explanation.
2. In the Materials and methods (line 150), the GFPs for this study were obtained from Jiangsu Shufeng Prawn Breeding Co LTD (Gaoyou, Jiangsu, China). In China, there are many nationally approved varieties of *Macrobrachium rosenbergii*. Could author please specify the variety of GFP used in this experiment and provide a description?
3. The GFPs in the Figure 3 is drawn by the authors. Could the author explain why real images of GFPs were not used?
4. In Results, this study identified numerous key genes associated with phenotypic differentiation, particularly those discovered through association models. If these genes can be assigned gene names via sequence alignment, it is preferable to use the gene names directly.
5. In Discussion, I believe the highlight of this study lies in its analysis of the correlations among host genes, gut microbiota, and phenotypic traits to systematically reveal the mechanisms underlying male phenotypic differentiation. It is suggested that the authors elaborate on this aspect in the discussion section to emphasize the study's originality and significance.

Other points:

1. Page 47, the line spacing of the title below Figure 8 is different from that of other figures. Please correct it.
2. Please delete page 37.
3. I suggest that the authors move Table 4 and Table 5 to the supplementary data.
4. In academic formatting, tables typically follow the three-line table style. Notably, the second line in both Table 4 and Table 5 appears noticeably thinner than the others. Please adjust this to ensure uniform line thickness for all three lines in the tables.
5. In References, the formats of many references are incorrect. Please correct them.

Reviewer #2 (Comments for the Author):

The manuscript by Zhao et al. focuses on the phenomenon of male morphotype differentiation in *Macrobrachium rosenbergii*. Through the study of the intestinal microbiota of male *Macrobrachium rosenbergii* at different ages and the integration of transcriptome data, this paper reveals the relationship between the intestinal microbiota, transcripts, and phenotypes of different morphotypes of *Macrobrachium rosenbergii*, and determines the molecular factors leading to the differentiation of male giant freshwater prawns (*Macrobrachium rosenbergii*) into different morphotypes. At present, there are few studies on the morphotype differentiation of male *Macrobrachium rosenbergii*, and this study on the morphotype differentiation phenomenon of male *Macrobrachium rosenbergii* at different ages is relatively novel. This research is conducive to a better understanding of the molecular mechanism of morphotype differentiation in male *Macrobrachium rosenbergii*. However, before this paper can be accepted for publication, the authors still need to address the following issues. The detailed information is as follows.

1. In the Introduction, it is recommended that the authors supplement information on the global production of *Macrobrachium rosenbergii* and its economic status in China, so as to highlight the significance of this study.
2. In the Method (Sample collection), line 160, the manuscript mentions "A total of 66 individuals (18 161 BCs, 18 OCs, and 30 SMs) were sampled". In this experiment, sampling was conducted at three time points. It is preferable for the authors to elaborate on the sampling quantity at each time point. Additionally, a brief explanation is needed regarding why the sampling number for SMs was more than that for other phenotypes (BCs and OCs).
3. In the Method (Phenotypic Trait Measurement), it is advisable for the authors to supplement the brand and specifications of the measurement tools.
4. In Figure 3, the text labels for the intestine in Figure 3A are incorrect. Both the midgut segment and the hindgut segment are labeled as "midgut", and the hindgut segment should be labeled as "hindgut." Please correct this. Additionally, please specify which segment of intestinal tissue was sampled in this study.
5. In the text, gene names should be presented in full at their first mention, with abbreviations used subsequently. However, many genes are consistently referred to by their full names throughout the text-this should be corrected (such as prophenoloxidase activating enzyme, PPA).
6. In the Materials and methods (line156), the manuscript mentions "special feed without probiotics", but the specific components of the feed (such as whether it contained antibiotics or natural plant extracts) were not specified. Some components may affect the growth of *Macrobrachium rosenbergii* and structure of the intestinal flora. If other feed additives were used, please provide a supplementary explanation.
7. Although 75% ethanol was mentioned for surface disinfection, the disinfection time and subsequent aseptic operation procedures (such as whether to rinse or not) were not elaborated in detail. It is suggested to supplement the specific parameters of the disinfection steps to eliminate the interference of exogenous bacteria.
8. The paper focuses on male differentiation but does not mention comparative data on the intestinal microbiota or gene expression of females. Although the research objective is clear, it is suggested that a brief explanation be given in the discussion as to why female controls were not included.

9. The gene names in the text were not uniformly in italic format (such as Wnt-6, CTSSB, TP53), and the protein names (such as "calpain clp-1") did not follow the rule of "capitalized first letter + non-italic". Please check the full text carefully.
10. Data availability link is missing, complete links.
11. The statistic "p" should be in lowercase and italic ($p < 0.05$).

Reviewer #3 (Comments for the Author):

This manuscript explores the tripartite interaction between host genes, gut microbiota, and phenotypic differentiation in male *Macrobrachium rosenbergii* (GFP). The subject matter is of notable interest for aquaculture genetics and microbiome research. However, several aspects of the manuscript require substantial revision to improve clarity, scientific rigor, and presentation.

Major Comments

Length and Focus of Key Sections

The Introduction and Discussion sections are disproportionately long and often redundant. The authors should condense these parts to maintain reader engagement and ensure focus on the main findings.

Additionally, the figures require substantial improvement in terms of visual clarity. All figures should be redesigned to ensure they are legible on publication scale. Furthermore, figure references within the main text should be concise and used judiciously.

Significant portions of this manuscript may have been generated or assisted by artificial intelligence, which is reflected in the writing style, particularly in the overly verbose and sometimes disjointed structure of certain sections.

Abstract Clarity

Lines 36-38 introduce redundant procedural detail that should be excluded from the abstract. The abstract should focus on concise outcomes and significance.

Scientific Accuracy and Language Corrections

Line 60: The statement regarding "disease susceptibility" is misleading-this study does not directly evaluate disease susceptibility, and this claim should be removed or clarified.

Line 62: This sentence needs rewriting for clarity and better integration into the argument.

Line 84-85: Reordering is suggested for logical flow. The sentence beginning with "Gut microbiota, often referred to as phenotype sculptors..." should precede the discussion of genotype-microbiota interaction.

Figure and Table Presentation

Figures 3 to 9 suffer from poor visibility due to small font sizes. These need to be reformatted for clarity.

Figure 2 resembles a review-type schematic and lacks experimental clarity; consider revising it to better represent your data or move to the Supplement.

Figure references for the Introduction (e.g., lines 100, 102) should be removed; figures should be reserved for Results.

Figures 5 and 8 are redundant given the text's genus-level focus; suggest moving the supplementary material.

Tables 4 and 5 (transcriptome statistics) are more appropriate for supplementary material.

Technical Clarifications

Line 162: The ethics section lacks the necessary approval code. Please include the institutional approval reference number.

The manuscript refers to "16S rDNA analysis"; the correct terminology in current standards is "16S rRNA gene analysis."

Clarify whether OTU-based or ASV-based (amplicon sequence variant) microbial classification was used. The field has largely moved to ASVs due to greater resolution.

Results Section Structure

Line 244: The subtitle used for this results subsection is vague and should be rewritten to reflect the key findings.

Lines 290-297 and 308: Ensure that relevant figures or tables are cited when referring to data.

Lines 314-317: This sentence is repeated and should be removed for conciseness.

Lines 365-370: Rewriting is needed to fit the formal tone of the Results. Avoid including raw GO term identifiers (e.g., GO:XXXX) in the main text-they reduce readability.

Transcriptome Data Reporting

The TRINITY IDs (e.g., TRINITY_DNXXXX) are too long and detract from readability. These can be abbreviated or referenced more concisely.

Presenting transcriptome results across three separate points inflates the manuscript unnecessarily; a more integrated summary would be more efficient.

Minor Comments

Proofreading is needed throughout for grammar, clarity, and conciseness.

Discussion sections, especially Lines 504-511, extensively describe methods, which are not appropriate for this section. This must be significantly shortened and revised to center on interpreting key results.

The nomenclature (e.g., gene and bacterial species) should be consistently italicized per journal standards.

Improve logical transitions between paragraphs in both Introduction and Discussion.

Recommendation

The manuscript has potential but requires major revisions to meet publication standards. Upon addressing the above points, particularly those related to clarity, figure quality, methodological precision, and structural conciseness, it may be reconsidered for publication.

The manuscript by Zhao et al. focuses on the phenomenon of male morphotype differentiation in *Macrobrachium rosenbergii*. Through the study of the intestinal microbiota of male *Macrobrachium rosenbergii* at different ages and the integration of transcriptome data, this paper reveals the relationship between the intestinal microbiota, transcripts, and phenotypes of different morphotypes of *Macrobrachium rosenbergii*, and determines the molecular factors leading to the differentiation of male giant freshwater prawns (*Macrobrachium rosenbergii*) into different morphotypes. At present, there are few studies on the morphotype differentiation of male *Macrobrachium rosenbergii*, and this study on the morphotype differentiation phenomenon of male *Macrobrachium rosenbergii* at different ages is relatively novel. This research is conducive to a better understanding of the molecular mechanism of morphotype differentiation in male *Macrobrachium rosenbergii*. However, before this paper can be accepted for publication, the authors still need to address the following issues. The detailed information is as follows.

1. In the Introduction, it is recommended that the authors supplement information on the global production of *Macrobrachium rosenbergii* and its economic status in China, so as to highlight the significance of this study.
2. In the Method (Sample collection), line 160, the manuscript mentions “A total of 66 individuals (18 BCs, 18 OCs, and 30 SMs) were sampled”. In this experiment, sampling was conducted at three time points. It is preferable for the authors to elaborate on the sampling quantity at each time point. Additionally, a brief explanation is needed regarding why the sampling number for SMs was more than that for other phenotypes (BCs and OCs).
3. In the Method (Phenotypic Trait Measurement), it is advisable for the authors to supplement the brand and specifications of the measurement tools.
4. In Figure 3, the text labels for the intestine in Figure 3A are incorrect. Both the midgut segment and the hindgut segment are labeled as "midgut", and the hindgut segment should be labeled as "hindgut." Please correct this. Additionally, please specify which segment of intestinal tissue was sampled in this study.
5. In the text, gene names should be presented in full at their first mention, with

abbreviations used subsequently. However, many genes are consistently referred to by their full names throughout the text—this should be corrected (such as prophenoloxidase activating enzyme, PPA).

6. In the Materials and methods (line156), the manuscript mentions “special feed without probiotics”, but the specific components of the feed (such as whether it contained antibiotics or natural plant extracts) were not specified. Some components may affect the growth of *Macrobrachium rosenbergii* and structure of the intestinal flora. If other feed additives were used, please provide a supplementary explanation.
7. Although 75% ethanol was mentioned for surface disinfection, the disinfection time and subsequent aseptic operation procedures (such as whether to rinse or not) were not elaborated in detail. It is suggested to supplement the specific parameters of the disinfection steps to eliminate the interference of exogenous bacteria.
8. The paper focuses on male differentiation but does not mention comparative data on the intestinal microbiota or gene expression of females. Although the research objective is clear, it is suggested that a brief explanation be given in the discussion as to why female controls were not included.
9. The gene names in the text were not uniformly in italic format (such as Wnt-6, CTSB, TP53), and the protein names (such as "calpain clp-1") did not follow the rule of "capitalized first letter + non-italic". Please check the full text carefully.
10. Data availability link is missing, complete links.
11. The statistic "p" should be in lowercase and italic ($p < 0.05$).

Response to reviewers

Manuscript Number: Spectrum01290-25 (Interaction of host gene-gut microbiota in male grading of *Macrobrachium rosenbergii*)

To Reviewer 1: This study identified host KEGG pathways and key genes associated with morphotype differentiation, and further conducted association analyses to elucidate the tripartite relationships among gut microbiota, phenotypic traits, and gene expression. The results revealed the molecular mechanisms underlying morphotype-specific phenotypic variation, providing critical insights for developmental studies and genetic improvement strategies aimed at optimizing the sustainability of GFP aquaculture. The findings of this study are novel and interesting, providing a new perspective for understanding the mechanism underlying male classification in GFPs. However, there are some issues that the authors still need to address before the paper is acceptable for publication. The detail information is as following:

Major points:

1. In Introduction, the authors state that male *Macrobrachium rosenbergii* is divided into three morphotypes (SM, OC and BC), but some studies have indicated that it can be divided into four morphotypes (SM, OC, BC and OBC). Please provide an explanation.

Response: We appreciate the reviewer's keen observation regarding morphotype classification. The reviewer is correct that some literature describes four distinct morphotypes: small male (SM), orange claw males (OC), blue claw males (BC), and old blue claw males (OBC). Our study primarily employs the three-morphotype system (SM, OC, BC). This classification is widely adopted and focuses on the primary, reproductively active stages with distinct morphological and behavioral characteristics driving social hierarchy formation. The OBC stage is often considered a terminal phase within the BC morphotype, characterized by senescence and reduced reproductive activity rather than representing a fundamentally distinct functional category in the social hierarchy dynamics central to our study.

2. In the Materials and methods (line 150), the GFPs for this study were obtained from Jiangsu Shufeng Prawn Breeding Co LTD (Gaoyou, Jiangsu, China). In China, there are many nationally approved varieties of *Macrobrachium rosenbergii*. Could author please specify the variety of GFP

used in this experiment and provide a description?

Response: We thank the reviewer for highlighting the need for specificity regarding the genetic background of the study animals. Upon confirmation with Jiangsu Shufeng Prawn Breeding Co. Ltd., the Giant Freshwater Prawns (GFPs) used in this experiment belong to the “Shufeng No.1” variety. This variety is a nationally certified improved strain in China, selectively bred for traits including rapid growth rate, uniform size, and enhanced disease resistance. We have amended the Materials and Methods section (Section 2.1, Line 147) to explicitly state:

Giant freshwater prawns (*Macrobrachium rosenbergii*) of the “Shufeng No.1” variety were obtained from Jiangsu Shufeng Prawn Breeding Co. LTD (Gaoyou, Jiangsu, China).

3. The GFPs in the Figure 3 is drawn by the authors. Could the author explain why real images of GFPs were not used?

Response: We appreciate the reviewer's query about the figure presentation. Figure 3 (schematic representation of morphotypes) was intentionally designed as a schematic illustration rather than using photographs for the following reasons:

Schematics allow us to precisely highlight the key diagnostic morphological features (e.g., relative claw size and shape, body proportions, spine development) that distinguish the SM, OC, and BC morphotypes in a consistent and unambiguous manner. Photographs can be variable due to lighting, angle, individual variation, and background clutter, potentially obscuring these critical features. The schematic enables us to exaggerate subtle but important differences (e.g., the distinct curvature of the OC claw vs. the BC claw) for immediate visual recognition by the reader, which might be less apparent in a typical photograph.

4. In Results, this study identified numerous key genes associated with phenotypic differentiation, particularly those discovered through association models. If these genes can be assigned gene names via sequence alignment, it is preferable to use the gene names directly.

Response: We agree with the reviewer that using recognized gene names enhances readability and biological context. We have made modifications to the relevant parts. However, the challenge lies in the fact that a large number of genes cannot be accurately named through sequence alignment, which is why we are also using codes for representation.

5. In Discussion, I believe the highlight of this study lies in its analysis of the correlations among host genes, gut microbiota, and phenotypic traits to systematically reveal the mechanisms underlying male phenotypic differentiation. It is suggested that the authors elaborate on this aspect in the discussion section to emphasize the study's originality and significance.

Response: We sincerely thank the reviewer for recognizing the core integrative contribution of our work. We agree that this multi-omics integration is the study's most significant and novel aspect. We have supplemented the following contents :

This study provides the first evidence in decapod crustaceans that male phenotypic differentiation arises from dynamic interactions between host genomic variation, gut microbiome assembly, and trait development. The application of tripartite association modeling (intestinal bacteria—phenotypic traits—DEG) in a non-model crustacean represents a significant technical advance. Such host-microbiome coevolutionary dynamics may generalize to other polymorphic species.

Other points:

1. Page 47, the line spacing of the title below Figure 8 is different from that of other figures. Please correct it.

Response: We have made the modifications.

2. Please delete page 37.

Response: We have deleted the blank page here.

3. I suggest that the authors move Table 4 and Table 5 to the supplementary data.

Response: We have made the modifications.

4. In academic formatting, tables typically follow the three-line table style. Notably, the second line in both Table 4 and Table 5 appears noticeably thinner than the others. Please adjust this to ensure uniform line thickness for all three lines in the tables.

Response: We have made the modifications.

5. In References, the formats of many references are incorrect. Please correct them.

Response: We have made the modifications.

Reviewer #2 (Comments for the Author):

The manuscript by Zhao et al. focuses on the phenomenon of male morphotype differentiation in *Macrobrachium rosenbergii*. Through the study of the intestinal microbiota of male *Macrobrachium rosenbergii* at different ages and the integration of transcriptome data, this paper reveals the relationship between the intestinal microbiota, transcripts, and phenotypes of different morphotypes of *Macrobrachium rosenbergii*, and determines the molecular factors leading to the differentiation of male giant freshwater prawns (*Macrobrachium rosenbergii*) into different morphotypes. At present, there are few studies on the morphotype differentiation of male *Macrobrachium rosenbergii*, and this study on the morphotype differentiation phenomenon of male *Macrobrachium rosenbergii* at different ages is relatively novel. This research is conducive to a better understanding of the molecular mechanism of morphotype differentiation in male *Macrobrachium rosenbergii*. However, before this paper can be accepted for publication, the authors still need to address the following issues. The detailed information is as follows.

1. In the Introduction, it is recommended that the authors supplement information on the global production of *Macrobrachium rosenbergii* and its economic status in China, so as to highlight the significance of this study.

Response: We supplemented the content.

In 2024, the global production of *Macrobrachium rosenbergii* is approximately 356,000 tons, with China's total output exceeding 196,000 tons (1).

1. FAO. 2024. The State of World Fisheries and Aquaculture – Blue Transformation in action. Rome.

2. In the Method (Sample collection), line 160, the manuscript mentions "A total of 66 individuals (18 161 BCs, 18 OCs, and 30 SMs) were sampled". In this experiment, sampling was conducted at

three time points. It is preferable for the authors to elaborate on the sampling quantity at each time point. Additionally, a brief explanation is needed regarding why the sampling number for SMs was more than that for other phenotypes (BCs and OCs).

Response: We are very grateful for the comments put forward by the reviewers. Due to our negligence and failure to explain clearly here, we have made the following modifications:

At each time point, we selected 66 individuals for sample collection (18 BCs, 18 OCs, and 30 SMs), and a total of 198 individuals were collected at the three time points. Due to the relatively small sample size of the SM-grade tissue, in order to ensure the required sample size for sampling, the number of SM samples was greater than that of the other phenotypes.

3. In the Method (Phenotypic Trait Measurement), it is advisable for the authors to supplement the brand and specifications of the measurement tools.

Response: An electronic balance (Mettler Toledo ME203E, precision: 0.01 g) and a digital Vernier caliper (Mitutoyo CD-15CX, precision: 0.01 mm) were used to measure 15 phenotypic traits:

4. In Figure 3, the text labels for the intestine in Figure 3A are incorrect. Both the midgut segment and the hindgut segment are labeled as "midgut", and the hindgut segment should be labeled as "hindgut." Please correct this. Additionally, please specify which segment of intestinal tissue was sampled in this study.

Response: We have made the modifications.

Whole intestine and gonadal samples were aseptically collected from SM, OC, and BC individuals using sterile scissors and forceps.

5. In the text, gene names should be presented in full at their first mention, with abbreviations used subsequently. However, many genes are consistently referred to by their full names throughout the text-this should be corrected (such as prophenoloxidase activating enzyme, PPA).

Response: We carefully examined and revised the text.

6. In the Materials and methods (line156), the manuscript mentions "special feed without

probiotics", but the specific components of the feed (such as whether it contained antibiotics or natural plant extracts) were not specified. Some components may affect the growth of *Macrobrachium rosenbergii* and structure of the intestinal flora. If other feed additives were used, please provide a supplementary explanation.

Response: The GFPs were fed a specialized GFP diet provided by Jiangsu Fuyuda Grain Products Co., Ltd. Special feed contained no probiotics or antibiotics. The main components of the feed are crude protein $\geq 38\%$, crude fat $\geq 5\%$, and crude fiber $\leq 7\%$.

7. Although 75% ethanol was mentioned for surface disinfection, the disinfection time and subsequent aseptic operation procedures (such as whether to rinse or not) were not elaborated in detail. It is suggested to supplement the specific parameters of the disinfection steps to eliminate the interference of exogenous bacteria.

Response: We have made the modifications.

To minimize exogenous bacterial contamination, each individual was immersed in 75% ethanol for 30 seconds and then rinsed three times with sterile PBS.

8. The paper focuses on male differentiation but does not mention comparative data on the intestinal microbiota or gene expression of females. Although the research objective is clear, it is suggested that a brief explanation be given in the discussion as to why female controls were not included.

Response: We thank the reviewer for this pertinent observation. The exclusion of female *Macrobrachium rosenbergii* from this study was deliberate and scientifically justified. Morphotype differentiation (SM/OC/BC) is a male-exclusive phenomenon in *M. rosenbergii*, driven by androgenic gland hormones and social hierarchy dynamics. Females exhibit no equivalent morphological polymorphism (e.g., lack claw dimorphism, uniform growth patterns) and do not participate in the dominance behaviors central to this study. Furthermore, since our research focus is on male *M. rosenbergii*, including female *M. rosenbergii* in the discussion does not conform to the logic mentioned earlier. Therefore, we did not include this part in the discussion.

9. The gene names in the text were not uniformly in italic format (such as Wnt-6, CTSB, TP53), and the protein names (such as "calpain clp-1") did not follow the rule of "capitalized first letter + non-italic". Please check the full text carefully.

Response: We have made the modifications.

10. Data availability link is missing, complete links.

Response: We have made the modifications.

The raw sequence data by illumina sequencing platform (SRR33015805-SRR33015858) have been deposited in the NCBI Sequence Read Archive (SRA) database under BioProject number PRJNA1247755 (URL: <https://www.ncbi.nlm.nih.gov/sra/PRJNA1247755>). RNA-seq data (SRR33048285-SRR33048338) has been uploaded to NCBI SRA database under BioProject number PRJNA1248510 (URL: <https://www.ncbi.nlm.nih.gov/sra/PRJNA1248510>).

11. The statistic "p" should be in lowercase and italic ($p < 0.05$).

Response: We carefully examined and revised the text.

Reviewer #3 (Comments for the Author):

This manuscript explores the tripartite interaction between host genes, gut microbiota, and phenotypic differentiation in male *Macrobrachium rosenbergii* (GFP). The subject matter is of notable interest for aquaculture genetics and microbiome research. However, several aspects of the manuscript require substantial revision to improve clarity, scientific rigor, and presentation.

Major Comments

Length and Focus of Key Sections

The Introduction and Discussion sections are disproportionately long and often redundant. The authors should condense these parts to maintain reader engagement and ensure focus on the main findings.

Additionally, the figures require substantial improvement in terms of visual clarity. All figures

should be redesigned to ensure they are legible on publication scale. Furthermore, figure references within the main text should be concise and used judiciously.

Significant portions of this manuscript may have been generated or assisted by artificial intelligence, which is reflected in the writing style, particularly in the overly verbose and sometimes disjointed structure of certain sections.

Response: We sincerely appreciate the constructive feedback on our manuscript. We have carefully addressed all comments through substantial revisions to improve clarity, scientific rigor, and presentation. Key changes are summarized below and marked in the revised manuscript.

Abstract Clarity

Lines 36-38 introduce redundant procedural detail that should be excluded from the abstract. The abstract should focus on concise outcomes and significance.

Response: We have made the modifications.

The interactions among phenotypic traits, gut microbiota and key genes were systematically studied through association analysis.

Scientific Accuracy and Language Corrections

Line 60: The statement regarding "disease susceptibility" is misleading-this study does not directly evaluate disease susceptibility, and this claim should be removed or clarified.

Response: We have made the modifications.

This species is known for its rapid growth and short cultivation cycle.

Line 62: This sentence needs rewriting for clarity and better integration into the argument.

Response: We have made the modifications.

Since its introduction to China in the 1970s, GFP aquaculture has expanded significantly, making China the largest global producer.

Line 84-85: Reordering is suggested for logical flow. The sentence beginning with "Gut microbiota, often referred to as phenotype sculptors..." should precede the discussion of genotype-microbiota interaction.

Response: We adjusted the paragraph structure.

Figure and Table Presentation

Figures 3 to 9 suffer from poor visibility due to small font sizes. These need to be reformatted for clarity.

Response: We have made every effort to ensure the images are presented in the highest possible clarity.

Figure 2 resembles a review-type schematic and lacks experimental clarity; consider revising it to better represent your data or move to the Supplement.

Response: We have moved it to the supplementary materials.

Figure references for the Introduction (e.g., lines 100, 102) should be removed; figures should be reserved for Results.

Response: This section of the images aims to provide readers with a more intuitive understanding of the grade differentiation phenomenon in male *Macrobrachium rosenbergii*. Therefore, we have retained some of the images.

Figures 5 and 8 are redundant given the text's genus-level focus; suggest moving the supplementary material.

Response: We have moved it to the supplementary materials.

Tables 4 and 5 (transcriptome statistics) are more appropriate for supplementary material.

Response: We have made the modifications.

Technical Clarifications

Line 162: The ethics section lacks the necessary approval code. Please include the institutional approval reference number.

Response: We have made the modifications.

All experimental procedures were carried out in accordance with the Chinese Guideline for Laboratory Animal Care and Use approved by the Animal Ethics Review Committee of Huzhou

University (No. HUZU-20230503, Huzhou, China).

The manuscript refers to "16S rDNA analysis"; the correct terminology in current standards is "16S rRNA gene analysis."

Response: We have made the modifications.

Clarify whether OTU-based or ASV-based (amplicon sequence variant) microbial classification was used. The field has largely moved to ASVs due to greater resolution.

Response: We have made the modifications.

Based on the operational taxonomic units (OTUs) microbial classification, these sequences were clustered into 755 OTUs at a 97% similarity threshold, with 22-323 OTU counts per sample.

Results Section Structure

Line 244: The subtitle used for this results subsection is vague and should be rewritten to reflect the key findings.

Response: We have made the modifications.

Phenotypic Trait Analysis of Male GFP Morphotypes at Various Developmental Stages

Lines 290-297 and 308: Ensure that relevant figures or tables are cited when referring to data.

Response: We conducted thorough checks on the data and made necessary modifications, referring to the table for validation.

Lines 314-317: This sentence is repeated and should be removed for conciseness.

Response: We have made the modifications.

At the species level, *L. garvieae* and *L. taiwanensis* being the most abundant (Fig. 7C).

Lines 365-370: Rewriting is needed to fit the formal tone of the Results. Avoid including raw GO term identifiers (e.g., GO:XXXX) in the main text-they reduce readability.

Response: We have made the modifications.

Transcriptome Data Reporting

The TRINITY IDs (e.g., TRINITY_DNXXXX) are too long and detract from readability. These can be abbreviated or referenced more concisely.

Response: We have made the modifications.

Presenting transcriptome results across three separate points inflates the manuscript unnecessarily; a more integrated summary would be more efficient.

Response: We sincerely appreciate the constructive feedback on our manuscript, we conducted an inspection and made modifications.

Minor Comments

Proofreading is needed throughout for grammar, clarity, and conciseness.

Discussion sections, especially Lines 504-511, extensively describe methods, which are not appropriate for this section. This must be significantly shortened and revised to center on interpreting key results.

Response: We have made the modifications.

Previous studies have identified potential DEGs, gut microbiota, and metabolites influencing male GFP morphotype differentiation. However, a systematic analysis of the developmental patterns and underlying mechanisms of male GFP morphotype differentiation remains lacking.

By integrating multi-omics data, we revealed the dynamic changes of phenotypic traits during the development of male GFPs and identified the key gut microbiota and potential functional genes.

The nomenclature (e.g., gene and bacterial species) should be consistently italicized per journal standards.

Response: We conducted a careful inspection and made modifications.

Improve logical transitions between paragraphs in both Introduction and Discussion.

Response: We have added logical transition phrases throughout the Introduction and Discussion sections to improve the flow of the narrative. This includes transitions between paragraphs

discussing background knowledge and our current study, as well as between different aspects of our findings in the Discussion."

Recommendation

The manuscript has potential but requires major revisions to meet publication standards. Upon addressing the above points, particularly those related to clarity, figure quality, methodological precision, and structural conciseness, it may be reconsidered for publication.

Response: We thank the reviewer for their rigorous critique, which significantly strengthened the manuscript. All revisions have been made to best comply with the journal's standards, and we believe the study now presents a clear, robust contribution to aquaculture genetics and microbiome research.

Re: Spectrum01290-25R1 (Interaction of host gene-gut microbiota in male grading of *Macrobrachium rosenbergii*)

Dear Prof. Quanxin Gao:

Your manuscript has been accepted, and I am forwarding it to the ASM production staff for publication. Your paper will first be checked to make sure all elements meet the technical requirements. ASM staff will contact you if anything needs to be revised before copyediting and production can begin. Otherwise, you will be notified when your proofs are ready to be viewed.

Sincerely,
John Chaston
Editor
Microbiology Spectrum

Reviewer #1 (Comments for the Author):

The author has made detailed modifications to the article based on the feedback.

Reviewer #3 (Comments for the Author):

No new comments on this edition